# Stopping Computation for Converged Tokens in Masked Diffusion-LM Decoding

**Daisuke Oba**[1][*]   **Danushka Bollegala**[2,3]   **Masahiro Kaneko**[4]   **Naoaki Okazaki**[1]
[1]Institute of Science Tokyo   [2]University of Liverpool   [3]Amazon   [4]MBZUAI

## Abstract

Masked Diffusion Language Models generate sequences via iterative sampling that progressively unmasks tokens. However, they still recompute the attention and feed-forward blocks for every token position at every step—even when many unmasked tokens are essentially fixed, resulting in substantial waste in compute. We propose **SureLock**: when the posterior at an unmasked position has stabilized across steps (our *sure* condition), we *lock* that position—thereafter skipping its query projection and feed-forward sublayers—while caching its attention keys and values so other positions can continue to attend to it. This reduces the dominant per-iteration computational cost from $O(N^2 d)$ to $O(MNd)$ where $N$ is the sequence length, $M$ is the number of unlocked token positions, and $d$ is the model dimension. In practice, $M$ decreases as the iteration progresses, yielding substantial savings. On LLaDA-8B, SureLock reduces algorithmic FLOPs by 30–50% relative to the same sampler without locking, while maintaining comparable generation quality. We also provide a theoretical analysis to justify the design rationale of SureLock: monitoring only the local KL at the lock step suffices to bound the deviation in final token probabilities. Our project page is available at https://daioba.github.io/surelock.

## 1 Introduction

Discrete diffusion language models (DLMs) generate text by iteratively denoising a discrete sequence over $T$ steps (Li et al., 2022; Schiff et al., 2024); While its formulation varies (e.g., token swap, insertion, or masking), a widely used family operates through *masking and unmasking*—Masked Diffusion Language Models (MDLMs) (Sahoo et al., 2024; Shi et al., 2024; Nie et al., 2025; Arriola et al., 2025). Unlike autoregressive (AR) decoding (Vaswani et al., 2017)—whose per-step compute naturally grows by one query token via a $KV$–cache—standard diffusion-style sampling repeatedly recomputes self-attention and per-token feed-forward sublayers for *all* $N$ token positions[1] at *every* step—even for tokens that have already been unmasked and considered to be stable. Hence, the per-block cost is dominated by computing the attention scores $QK^\top$ and applying them to $V$, i.e., $O(N^2 d)$ where $d$ is the model dimension. It leads to substantial waste in compute.

To address the computational challenges of DLM sampling, prior work has mainly progressed along two axes: *Temporal* approaches shrink the step count—e.g., parallel/dilated unmasking and staged or learned samplers—thereby requiring fewer refinement (Luxembourg et al., 2025; Israel et al., 2025; Wei et al., 2025); *Reuse* approaches reduce per-step compute by reusing or approximating $K/V$ vectors and partially updating hidden features across steps (Ma et al., 2025; Liu et al., 2025b; Wu et al., 2025a). These choices chiefly either reduce the step count $T$ or amortize work across steps by reusing intermediate states; they do not alter the within-step spatial granularity: each step still issues $N$ query rows, so the attention-dominated cost remains $O(N^2 d)$ even in late iterations.

Beyond reducing the step count $T$ or reusing $K/V$ across steps, we pursue a largely orthogonal axis: *permanently and monotonically deactivating token positions as the sampling unfolds*. We propose **SureLock**: once a token's posterior has stabilized, we *lock* that position—we cache $K/V$ vectors

---

[*]Corresponding author. Email: daisuke.oba@nlp.comp.isct.ac.jp
[1]In this paper, we use *position* to refer to the a fixed token slot $i$ in the length-$N$ sequence; a *position* exists even while its token remains masked.

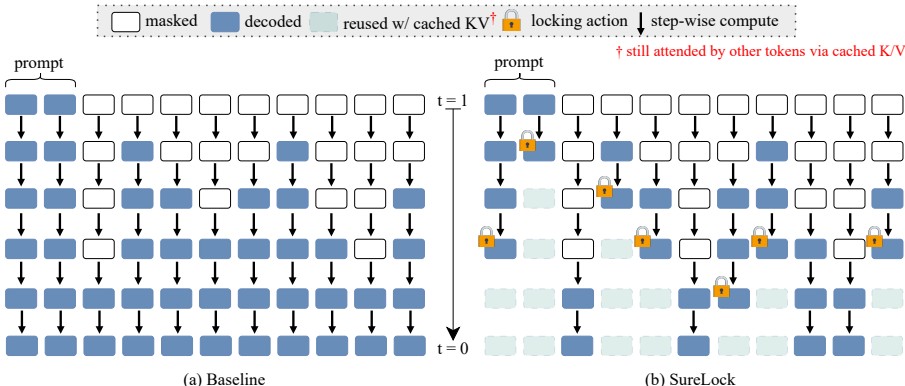

Figure 1: **Iterative sampling by a normal sampler and SureLock.** (*a*) Baseline consistently recomputes attention scores and FFN sublayers for every token position at every step even after the marginal tokens have become unmasked. (*b*) SureLock permanently stops recomputing for *locked* positions once these positions are locked. Via cached $K/V$, other tokens still attend to locked tokens.

and therefore *skip* its $Q$-projection and per-token feed-forward sublayers. Active token positions can still attend to the locked ones via the cached $K/V$ (Sec. 2). The per-block cost becomes $O(M_t N d)$ for attention and $O(M_t d^2)$ for FFN, instead of $O(N^2 d)$ and $O(N d^2)$, yielding a *monotonically decreasing* per-step compute profile as $M_t$ shrinks. We lock token positions based on their ***local stability***; a criterion whether the per-step KL divergence of the generative probability distribution falls below a threshold $\varepsilon$. In order to justify using KL as the locking signal, we derive a closed-form bound, which links a per-step KL at locking time to the terminal log-prob deviation[2]. This axis is *complementary* to the *Temporal* and *Reuse* approaches; existing methods continue to operate on the remaining active token positions.

Our work is most closely related to selection-based methods (e.g., DLLM-CACHE (Liu et al., 2025b)) lower per-step compute by updating only a subset of positions. SURELOCK, by contrast, answers a different question—***what to remove from compute permanently***, instead of *what to compute now*—so the active position set contracts over time; their selection target of step-wise compute can be tapered down monotonically.

We evaluate SURELOCK on representative MDLMs—LLaDA-8B-Base/-Instruct (Nie et al., 2025)—in Sec. 3. Across diverse decoding settings (e.g., sequence length, locking threshold), per-step algorithmic FLOPs are monotonically decreasing. On continuation generation with WikiText-103 (Merity et al., 2016) and instruction following with MT-Bench (Zheng et al., 2023), SURELOCK reduces algorithmic FLOPs up to **50%** in our runs without an expense of generation quality.

## 2 SURELOCK

We consider MDLMs that iteratively denoise a length-$N$ sequence, producing intermediate sequences over steps $t = 1, \ldots, T$. At step $t$, an $L$-block Transformer yields token-wise logits $z_t^{(i)}$ and posteriors $p_t^{(i)} = \mathrm{softmax}(z_t^{(i)})$ at token position $i$, together with hidden states $h_t^{(i)} \in \mathbb{R}^d$. Let $\mathcal{M}_t \subseteq [N]$ denote the masked positions (prediction targets) and $\bar{\mathcal{M}}_t = [N] \setminus \mathcal{M}_t$ the unmasked ones. In the standard dLLM sampler, each step recomputes self-attention and FFN for *all* tokens.

In the following, we explain the proposed method **SURELOCK** (Algorithm 1). Once a token's posterior has stabilized at a step, it locks that token position to stop step-wise computations at all subsequent steps by bypassing sublayers and caching K/V values (Sec. 2.1). Our locking criterion is based on the step-wise KL divergence of the posterior (Sec. 2.2). We explain the rationale of using the KL as locking criterion from a theoretical perspective (Sec. 2.3).

---

[2]It is deliberately conservative and is intended as a *design rationale*, not as a calibrated predictor for some specific models and datasets

---

**Algorithm 1** SureLock

---

**Require:** sequence length $N$, total steps $T$, confidence threshold percentile $m$, KL threshold $\varepsilon$
**Require:** vocabulary set $\mathcal{V}$, number of layers $L$, an unmasking policy UpdateMASK($\cdot$)
1: **State:** boolean lock $\in \{0,1\}^N$ (init: all 0), caches $\mathcal{C}$ for $(k,v)$ (init: None)
2: **State:** previous posteriors $p_{t-1}$ (init: None), masked indices $\mathcal{M}_t$ (init: $[N]$)
3: **State:** frozen block-input $\hat{x} \in \mathbb{R}^{N \times d}$ (init: prompt embeddings)
4: **Notation:** $X_t \in \mathbb{R}^{N \times d}$ denotes the model input at $t$ (from embeddings or previous step output)
5: **for** $t = 1$ **to** $T$ **do**                                                     ▷ one diffusion step
6:     $\mathcal{A}_t \leftarrow \{i \mid \text{lock}_i = 0\}, \quad \mathcal{L}_t \leftarrow \{i \mid \text{lock}_i = 1\}$          ▷ active and locked position at $t$
7:     **Initialize block input** $x^{(1)}[\mathcal{A}_t] \leftarrow X_t[\mathcal{A}_t], x^{(1)}[\mathcal{L}_t] \leftarrow \hat{x}[\mathcal{L}_t]$
8:     **Compute on active positions** $i \in \mathcal{A}_t$:
9:         **for** $\ell = 1$ **to** $L$
10:           $Q[\mathcal{A}_t], K[\mathcal{A}_t], V[\mathcal{A}_t] \leftarrow \text{Proj}_{Q,K,V}(\text{LN}(x^{(l)}[\mathcal{A}_t]))$          ▷ main projections
11:           $K^{\text{all}}[\mathcal{A}_t] \leftarrow K[\mathcal{A}_t], \ K^{\text{all}}[\mathcal{L}_t] \leftarrow \mathcal{C}.k[\mathcal{L}_t];$ **same for** $V^{\text{all}}$          ▷ assemble
12:           $h^{(l)}[\mathcal{A}_t] \leftarrow \text{FFN}(\text{Attn}(Q[\mathcal{A}_t], K^{\text{all}}, V^{\text{all}}))$          ▷ attention with FFN
13:           $x^{(l+1)}[\mathcal{A}_t] \leftarrow h^{(l)}[\mathcal{A}_t], \quad x^{(l+1)}[\mathcal{L}_t] \leftarrow x^{(l)}[\mathcal{L}_t]$          ▷ locked rows pass through
14:     **end for**
15:     **Obtain posteriors** $p_t[\mathcal{A}_t] \leftarrow \text{softmax}(\text{Proj}_{out}(x^{(L+1)}[\mathcal{A}_t]))$
16:     **Unmasking:** $\mathcal{M}_t \leftarrow \text{UpdateMASK}(p_t, \mathcal{M}_t), \quad \bar{\mathcal{M}}_t \leftarrow [N] \setminus \mathcal{M}_t$ ▷ unmask with posterior
17:     **Score on active positions** $i \in \mathcal{A}_t$:
18:         $u_t^{(i)} \leftarrow 1 - \max_{v \in \mathcal{V}} p_t^{(i)}(v)$ for $i \in \mathcal{A}_t$          ▷ confidence value
19:         $D_t^{(i)} \leftarrow \text{KL}(p_t^{(i)} \| p_{t-1}^{(i)})$ **if** $t > 1$ **else** $\infty$          ▷ step-wise KL
20:     **Locking candidates:** $\mathcal{Z}_t \leftarrow \mathcal{A}_t \cap \bar{\mathcal{M}}_t$          ▷ must be active & unmasked
21:         $\theta_m \leftarrow \text{Percentile}(\{u_t^{(j)}\}_{j \in \mathcal{Z}_t}, m)$          ▷ threshold over candidate positions
22:     **Lock:**
23:         $\mathcal{F}_t \leftarrow \{ i \in \mathcal{Z}_t \mid u_t^{(i)} \le \theta_m \wedge D_t^{(i)} \le \varepsilon \}$          ▷ locking evaluation
24:         $\text{lock}[\mathcal{F}_t] \leftarrow 1; \ \mathcal{C}_\ell.\{K,V\}[\mathcal{F}_t] \leftarrow \{K_\ell, V_\ell\}[\mathcal{F}_t] \ \forall \ell$ ▷ update locked indices and cache
25:         $\hat{x}[\mathcal{F}_t] \leftarrow x^{(1)}[\mathcal{F}_t]$          ▷ freeze block-input for future steps
26: **end for**

---

## 2.1 PERMANENTLY STOPPING STEP-WISE COMPUTE AND CACHING K/V

Once a token $i$ is deemed *converged* at step $t^\star$, we *permanently eliminate* its position from per-step compute: we **cache** its $K/V$ vectors, and **bypass** its query projection and per-token FFN at all subsequent steps. Let $\mathcal{L}_t$ be the set of indices locked by step $t$, and define the *active* set $\mathcal{A}_t :=$ $\bar{\mathcal{M}}_t \setminus \mathcal{L}_t$ (unmasked and not yet locked). At step $t$ we assemble the queries $Q[\mathcal{A}_t]$ and run a *variable-length* attention kernel against the full key/value tables $K^{\text{all}}$ and $V^{\text{all}}$ where $K^{\text{all}}[\mathcal{L}_t], V^{\text{all}}[\mathcal{L}_t]$ are read from cache. This yields attention outputs only for active positions $\in \mathcal{A}_t$, and we apply the FFN sublayers only to them. Moreover, for locked indices $i \in \mathcal{L}_t$ we keep their predictions fixed, i.e., $p_t^{(i)} \leftarrow p_{t^\star}^{(i)}$, and skip their FFN computation. Note that locked positions continue to be attended by other tokens via cached $K/V$.

## 2.2 CRITERION FOR LOCKING: STEP-WISE KL DIVERGENCE

Our locking rule is driven primarily by *local KL*; we will justify the validity of this design in Theorem 1 in Sec. 2.3. We also apply a *confidence* gating as a secondary safeguard to prefer more confident tokens with peaked posteriors. Disabling the confidence gate leaves the theorem unchanged.

**Primary: Local KL divergence.** For position $i$ at step $t$, we define the one-step divergence as

$$D_t^{(i)} \triangleq \text{KL}\left(p_t^{(i)} \| p_{t-1}^{(i)}\right).$$

**Optional: Confidence gate.** Let uncertainty $u_t^{(i)} = 1 - \max_{v \in \mathcal{V}} p_t^{(i)}(v)$ and $q_m(u_t)$ the empirical $m$-th percentile of $\{u_t^{(j)} : j \in \mathcal{A}_t\}$. When enabled, the gate accepts token position $i$ as a locking candidate, if $u_t^{(i)} \le q_m(u_t)$, i.e., top-$m\%$ confidence among active tokens.

**Locking rule.** We lock token position $i$ at step $t^\star$, if

$$D_{t^\star}^{(i)} \leq \varepsilon \quad \text{and, if the confidence gate is enabled,} \quad u_{t^\star}^{(i)} \leq q_m(u_{t^\star}).$$

The primary criterion alone suffices Theorem 1. Upon locking we cache $(k_{t^\star}^{(i)}, v_{t^\star}^{(i)})$ and remove position $i$ from $\mathcal{A}_{>t}$ thereafter. Unless otherwise noted, the criterion is based on the raw posterior before any temperature scaling. Therefore, SureLock is independent of sampling temperature in the categorical sampling (See appendix F for discussion on this choice).

## 2.3 DESIGN JUSTIFICATION

The purpose of this section is *design-theoretic*: we justify the use of a local KL threshold as a locking (freezing) criterion by proving that it upper-bounds the terminal error of the log-probability in closed form. The controller based on SURELOCK locks position $i$ at step $t^\star$, bypassing Q-projection and FNN sublayers, and reusing its cached $K/V$ thereafter (Alg. 1). We establish that the resulting error of the terminal log-probabilities, relative to an alternative identical sampler without locking, is bounded by $\delta = C_{\text{tail}}\sqrt{\varepsilon}$ under some assumptions, where $C_{\text{tail}}$ depends on operator-norm constants of the model and $\varepsilon$ is the threshold for determining lock (i.e., $D_{t^\star}^{(i)} \leq \varepsilon$). Therefore, the local KL criterion immediately converts to an explicit design for the terminal error ($\varepsilon(\delta) = \delta^2/C_{\text{tail}}^2$). In the following, $p_t^{(i)}$ and $z_t^{(i)}$ denote the posterior and logits at position $i$ after step $t$ in the no-lock run, while $\hat{p}_t^{(i)}$ denotes the corresponding posterior when $i$ is locked at $t^\star$.

*Note.* We emphasize that we here justify the use of a local KL threshold as a locking criterion by proving that it upper-bounds the terminal error in closed form, rather than to predict an empirical error value for a given specific setting.

**Standing assumptions for Theorem 1.** Theorem 1 relies on four main assumptions. We emphasize that these are simplifying regularity conditions introduced for analytical tractability, and that Theorem 1 should be interpreted as an idealized model of the behavior we observe empirically. For Assumption A2, which may appear rather strong, Appendix C shows that its implications do not deviate substantially from the practical situations. Let $z$ the logit and $f(z) = \log \text{softmax}(z)$. Fix constants $L > 0$, $L_{\text{sm}} > 0$, and $\rho \in (0, 1)$. For any position $i$ and step $s$, the following hold:

(A1) **Locking semantics.** Once $i$ is locked at $t^\star$, it is excluded from subsequent re-masking.

(A2) **Geometric tail contraction.** $D_s^{(i)} \leq \rho\, D_{s-1}^{(i)}$ for $s > t^\star$.

(A3) **One-step logit smoothness.** $\|z_s^{(i)} - z_{s-1}^{(i)}\|_2 \leq L\sqrt{D_{s-1}^{(i)}}$ *(derivation in Appendix D)*.

(A4) **Log-softmax Lipschitzness.** $\|f(z) - f(z')\|_\infty \leq L_{\text{sm}}\|z - z'\|_2$.

**Theorem 1** (Locking error bound and closed-form threshold)**.** *Fix a position $i$ that is unlocked up to $t^\star$ and then locked (Alg. 1). Under (A1)–(A4), for any terminal time $T > t^\star$,*

$$\left\|\log p_T^{(i)} - \log \hat{p}_T^{(i)}\right\|_\infty \leq C_{\text{tail}}\sqrt{D_{t^\star}^{(i)}}, \qquad C_{\text{tail}} := L_{\text{sm}}\, L/(1 - \sqrt{\rho}).$$

*In particular, if the locking test enforces $D_{t^\star}^{(i)} \leq \varepsilon$, then the terminal log-probability error is at most $\delta = C_{\text{tail}}\sqrt{\varepsilon}$, so the closed-form threshold is*

$$\varepsilon(\delta) = \delta^2/C_{\text{tail}}^2.$$

*Proof.* By (A1), locking token position $i$ at $t^\star$ permanently stops the $i$'s step-wise compute, so for all $t \geq t^\star$ we have $\hat{p}_t^{(i)} = p_{t^\star}^{(i)}$ and $\log \hat{p}_T^{(i)} = \log p_{t^\star}^{(i)}$. Therefore

$$\left\|\log p_T^{(i)} - \log \hat{p}_T^{(i)}\right\|_\infty = \left\|\log p_T^{(i)} - \log p_{t^\star}^{(i)}\right\|_\infty = \left\|f(z_T^{(i)}) - f(z_{t^\star}^{(i)})\right\|_\infty,$$

By the triangle inequality and telescoping over $s = t^\star + 1, \ldots, T$,

$$\|z_T^{(i)} - z_{t^\star}^{(i)}\|_2 \leq \sum_{s > t^\star}^{T} \|z_s^{(i)} - z_{s-1}^{(i)}\|_2.$$

Applying (A3) term-wise yields

$$\|z_T^{(i)} - z_{t^\star}^{(i)}\|_2 \ \le \ L \sum_{s=t^\star+1}^{T} \sqrt{D_{s-1}^{(i)}}.$$

Under (A2), $D_{s-1}^{(i)} \le \rho^{\,s-1-t^\star} D_{t^\star}^{(i)}$, therefore $\sqrt{D_{s-1}^{(i)}} \le \rho^{\frac{s-1-t^\star}{2}} \sqrt{D_{t^\star}^{(i)}}$ and

$$\sum_{s>t^\star} \sqrt{D_{s-1}^{(i)}} \le \frac{1}{1-\sqrt{\rho}} \sqrt{D_{t^\star}^{(i)}}.$$

Finally, by (A4):

$$\left\|\log p_T^{(i)} - \log p_{t^\star}^{(i)}\right\|_\infty \le L_{\text{sm}}\|z_T^{(i)} - z_{t^\star}^{(i)}\|_2 \le C_{\text{tail}}\sqrt{D_{t^\star}^{(i)}}$$

which proves the claim and the stated $\varepsilon(\delta)$. $\qquad\square$

## 2.4 Computational Complexity: Algorithmic FLOPs

This paper reports *algorithmic* FLOPs, counting only GEMMs (Q/K/V/Out projections, $QK^\top$ and $AV$, and FFN sublayers).[3] Let $N_{\text{gen}}$ the number of positions reserved for generation at initial step, $N_{\text{prompt}}$ the number of tokens for the prompt, $N_b := \max_{b\in B}(N_{\text{prompt}})+N_{\text{gen}}$ be the sequence length of batch $b$; $S$ the number of sampling steps; $B$ the batch size; $d$ the model dimension; $L$ the number of layers; $H$ the number of attention heads; $d_h := d/H$ the head size; $H_{\text{kv}}$ the number of $K/V$ heads; $d_{\text{ff}}$ the FFN dimension; $T_b := BN_b$ the number of token positions; $\mathcal{A}_{t,b}$ the active index set at $t$; and $M_{t,b} := |\mathcal{A}_{t,b}|$. We count algorithmic FLOPs as follows:

**Baseline.** For a step $t$, algorithmic FLOPs per batch are constant across $t$:

$$\mathcal{F}_{\text{base,b}}^t = L\,(\underbrace{4BHN_b^{\,2}d_h}_{QK^\top+AV} + \underbrace{2BN_bd^2}_{Q} + \underbrace{2BN_bd^2}_{\text{Out}} + \underbrace{4BN_bd\,H_{kv}d_h}_{K,V} + \underbrace{6BN_bd\,d_{\text{ff}}}_{\text{FFN}}).$$

**SureLock.** Algorithmic FLOPs exhibit temporal dynamics since step-wise computation is only for active positions $\in \mathcal{A}_{t,b}$, which changes across $t$:

$$\mathcal{F}_{\text{prop,b}}^t = \frac{|\mathcal{A}_{t,b}|}{BN_b}\mathcal{F}_{\text{base,b}}^t = \frac{M_{t,b}}{T_b}\mathcal{F}_{\text{base,b}}^t$$

## 3 Experiments

This section reports our empirical evaluation. We consider two kinds of basic tasks—language modeling and instruction following. In each run with different settings, profiling the per-batch sequence length $N_b$ and the number of active positions $M_{t,b} = |\mathcal{A}_{t,b}|$ at each step, we report computational complexity (algorithmic FLOPs). We also assess any quality degradation in these tasks induced by locking behavior of SURELOCK. Unless otherwise noted, we set the number of sampling steps to $S = N_{\text{gen}}$, and fix the percentile for the optional confidence gate (Sec. 2.2) to $m{=}20\%$. The block length for the semi-aggressive generation option is set to $N_g = N_{\text{gen}}$, yielding a fully parallel configuration. Temperature and classifier-free guidance scale are set to 0 by default.

### 3.1 Masked Diffusion Language Models and Datasets.

We evaluate leading open-weight and representative MDLMs with 8 billion parameters: LLaDA-8B-Base and LLaDA-8B-Instruct (Nie et al., 2025).

We utilize the following datasets: WikiText-103 (Merity et al., 2016) for language modeling and MT-Bench (Zheng et al., 2023) for instruction-following benchmarking. These benchmarks evaluate

---

[3]Element-wise operations (activations, gating, LN, softmax, RoPE) and cache I/O/packing are lower-order relative to the dominant $O(N_b^2 d)$ attention and $O(N_b d^2)$ FFN matmuls and occur similarly across methods, so omitting them does not affect relative comparisons.

| $N_{\text{gen}}$ | $\varepsilon$ | $\downarrow \bar{r}$ | $\downarrow \mathcal{F}_{\text{base}}$ | $\downarrow \mathcal{F}_{\text{prop}}$ | $\downarrow \mathcal{F}_{\text{prop}}/\mathcal{F}_{\text{base}}$ |
|---|---|---|---|---|---|
| 64 | 5e-4 | .548 | 9.017e+11 | 4.936e+11 | 0.547× |
| 64 | 5e-3 | .506 | 9.017e+11 | 4.565e+11 | 0.506× |
| 64 | 5e-2 | .482 | 9.017e+11 | 4.342e+11 | 0.482× |
| 256 | 5e-4 | .496 | 3.629e+12 | 1.801e+12 | 0.496× |
| 256 | 5e-3 | .482 | 3.629e+12 | 1.750e+12 | 0.482× |
| 256 | 5e-2 | .475 | 3.629e+12 | 1.725e+12 | 0.475× |
| 1024 | 5e-4 | .494 | 1.492e+13 | 7.366e+12 | 0.494× |
| 1024 | 5e-3 | .492 | 1.492e+13 | 7.333e+12 | 0.492× |
| 1024 | 5e-2 | .490 | 1.492e+13 | 7.315e+12 | 0.490× |

Table 1: **Algorithmic FLOPs**. $\bar{r}$ is the micro-averaged ratio of the number of active token positions.

complementary aspects: WIKITEXT-103 stresses core token-level modeling on broad, category-agnostic text, whereas MT-BENCH evaluates instruction-following ability and response helpfulness across eight categories (e.g., "writing", "coding"); Together, they cover both core modeling capacity and practical instruction-following behavior. Data usage settings vary by experimental set, so we explain them in the following as needed. Detailed settings are in Appendices G– I.

## 3.2 RESULTS

**Experiment-1: Computational Complexity.** We first evaluate the impact of SURELOCK on the reduction of algorithmic FLOPs, using LLaDA-8B-Instruct on a fixed set of 32 single-turn prompts from MT-Bench, which we sampled uniformly from each category (Appendix. G). We use a subset of MT-BENCH to efficiently sweep diverse inference settings. We set batch size $B = 4$, $N_{\text{gen}} = \{64, 256, 1024\}$, $\varepsilon = \{5e-4, 5e-3, 5e-2\}$, and run to record the actual number of active token positions $M_{t,b} = |\mathcal{A}_{t,b}|$ sequence length $N_b$ per-batch per-step. Table 1 shows that the computational complexity steadily decreased compared to the same sampler without applying SURELOCK. The smaller the locking criterion threshold $\varepsilon$, the lower the reduction rate; it aligns with our design intent where $\varepsilon$ is set for tightening the locking test (Sec. 2.2). Moreover, on the scale from 5e-2 to 5e-4, $\varepsilon$ is less influential to the reduction of computational complexity. We also report the micro-averaged ratio of the number of active positions; $\bar{r} = \sum_b \sum_t M_{t,b} / \sum_b \sum_t B N_b$ just as a reference to intuitively grasp how many active rows have been reduced. We can see that $\bar{r}$ is roughly aligned with the reduction ratio of algorithm FLOPs, suggesting that reducing the number of computable token positions leads to reducing computational load.

**Experiment-2: Temporal Dynamics of Computational Complexity.** Figure 2 shows the dynamics in computational complexity associated with the progression of sampling steps; logging data points was obtained from the same runs as the experiment-1. We can see that, as sampling progresses, the ratio of algorithmic FLOPs decreases at an accelerating rate. This is probably because surrounding most tokens are unmasked in the later steps ($M_t << N$) leading to the stability of local-KL. We put the results on the runs with different settings for $m=\{10, 40\}\%$ in the Appendix H.

**Experiment-3: Trade-off between Efficiency and Generation Quality.** We evaluate LLaDA-8B-Base on Wikitext-103, where we prompt a fixed set of text fragments, requiring the model to generate the continuation, and report Gen.-PPL measured by an external AR model—LLaMA-3-8B (Grattafiori et al., 2024). More details are in Appendix I. We also evaluate LLaDA-8B-Instruct on MT-Bench with single-turn settings; we report the averaged LLM-as-a-judge score from gpt-4o for the generated responses. Other details are described in Appendix I. We set batch size $B = 4$, $N_{\text{gen}} = \{64, 128, 256, 512\}$, $\varepsilon = \{5e-4, 5e-3\}$ throughout datasets, and $N_{\text{p}} = 64$ in WikiText-103. Table 2 and Table 3 show that while computational complexity is indeed decreasing, performance is being maintained at competitive levels. For the instruct model, the MT-Bench score is essentially unchanged across all settings (at most $-0.1$pt), despite the reduced compute. The base model follows the same trend in most configurations, but exhibits relatively larger degradation (up to $\geq 1.21\times$ Gen.-PPL) for short generation lengths ($N_{\text{gen}} \in \{64, 128\}$). It suggests that, in language modeling where prompts are likely to be open-ended, choosing very short $N_{\text{gen}}$ can lead to less desirable behavior under SURELOCK. While this can affect creative tasks like novel generation, such tasks are

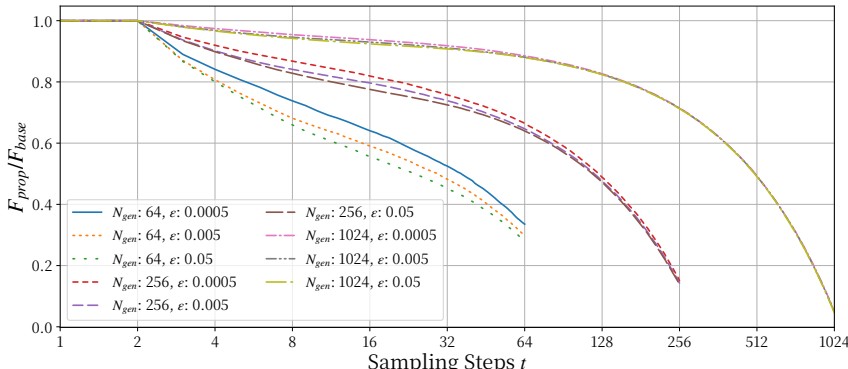

Figure 2: **Step-wise FLOPs ratio.** Ratio of step-wise algorithmic FLOPs, i.e., $\mathcal{F}^t_{\text{prop}}/\mathcal{F}^t_{\text{base}}$, consistently decreases as steps proceed, explaining later-step savings of computational cost.

| $N_{\text{gen}}$ | steps | $\varepsilon$ | $\downarrow$ Gen.-PPL$_{base}$ | $\downarrow \mathcal{F}_{base}$ | $\downarrow$ Gen.-PPL$_{prop}$ | $\downarrow \mathcal{F}_{prop}$ |
|---|---|---|---|---|---|---|
| 64 | 32 | 5e-4 | 5.7537 | 4.488e+11 | 6.0782 (1.06×) | 3.138e+11 (0.70×) |
| 64 | 32 | 5e-3 | 5.7537 | 4.488e+11 | 6.2177 (1.08×) | 2.788e+11 (0.62×) |
| 64 | 64 | 5e-4 | 3.4813 | 8.976e+11 | 4.5722 (1.31×) | 5.203e+11 (0.58×) |
| 64 | 64 | 5e-3 | 3.4813 | 8.976e+11 | 4.7596 (1.37×) | 4.664e+11 (0.52×) |
| 128 | 128 | 5e-4 | 2.6006 | 1.799e+12 | 2.9202 (1.12×) | 9.585e+11 (0.53×) |
| 128 | 128 | 5e-3 | 2.6006 | 1.799e+12 | 3.1042 (1.19×) | 8.964e+11 (0.50×) |
| 128 | 64 | 5e-4 | 3.1604 | 8.997e+11 | 3.8371 (1.21×) | 5.620e+11 (0.63×) |
| 128 | 64 | 5e-3 | 3.1604 | 8.997e+11 | 3.7353 (1.18×) | 5.083e+11 (0.57×) |
| 256 | 128 | 5e-4 | 2.3428 | 1.808e+12 | 2.6092 (1.11×) | 1.012e+12 (0.56×) |
| 256 | 128 | 5e-3 | 2.3428 | 1.808e+12 | 2.6687 (1.14×) | 9.497e+11 (0.53×) |
| 256 | 256 | 5e-4 | 2.0127 | 3.616e+12 | 2.0486 (1.02×) | 1.845e+12 (0.51×) |
| 256 | 256 | 5e-3 | 2.0127 | 3.616e+12 | 2.2103 (1.10×) | 1.779e+12 (0.49×) |
| 512 | 256 | 5e-4 | 1.6293 | 3.650e+12 | 1.7134 (1.05×) | 1.919e+12 (0.53×) |
| 512 | 256 | 5e-3 | 1.6293 | 3.650e+12 | 1.7526 (1.08×) | 1.851e+12 (0.51×) |
| 512 | 512 | 5e-4 | 1.4658 | 7.301e+12 | 1.5248 (1.04×) | 3.643e+12 (0.50×) |
| 512 | 512 | 5e-3 | 1.4658 | 7.301e+12 | 1.5444 (1.05×) | 3.588e+12 (0.49×) |

Table 2: Generation quality of continuation sequences with and without SURELOCK on LLaDA-8B-Base using WikiText-103. Gen.-PPL is the micro averaged PPL evaluated with LLaMA-3-8B.

rarely run with very short outputs. As an additional check for a more demanding downstream task, we also conduct a small-scale evaluation of LLaDA-8B-Instruct on the HumanEval code-generation benchmark (Chen et al., 2021). Here, correctness is assessed by executing the generated code against unit tests, where even small local errors can cause failure. Under a representative decoding configuration, SURELOCK shows no deterioration in Pass@1 while providing substantial compute reduction, i.e., $\sim 0.5x$ (Appendix J), indicating that the observed changes in Gen.-PPL largely correspond to benign surface-level variations that do not break sentence structure or semantics. We also provide different analyses in Experiment-6 on how to interpret Gen.-PPL.

**Experiment-4: Runtime Performance.** We evaluate End-to-end Token/Sec (E2E-TPS), which is the sustained decoding throughput of the model across multiple batches, and temporal dynamics for step-wise TPS. See Appendix K for more details on the metrics. We follow the settings in Experiment-1 for the dataset and the model. We set batch size $B = \{1, 2, 4, 8\}$, $N_{\text{gen}} = \{64, 256, 1024\}$, $\varepsilon = 5e-3$. Figure 3a shows that SURELOCK can achieve greater runtime gains in compute-bound areas e.g., $N \geq 256$, $B > 1$. However, no runtime gain was observed under relatively light computational load settings ($N_g = 64$, $B = 1$); this differs from the trend in computational complexity. By design, SureLock replaces dense, uniform computation with more irregular token-sparse computation, so some implementation-level overheads (e.g., irregular cache accesses) can offset part of the FLOP savings, especially for smaller or moderately sized models. As a result,

| $N_{\text{gen}}$ | steps | $\varepsilon$ | $\uparrow$ Score$_{base}$ | $\downarrow \mathcal{F}_{base}$ | $\uparrow$ Score$_{prop}$ | $\downarrow \mathcal{F}_{prop}$ |
|---|---|---|---|---|---|---|
| 64 | 32 | 5e-4 | 3.9 | 4.515e+11 | 3.8 (0.97×) | 3.153e+11 (0.698×) |
| 64 | 32 | 5e-3 | 3.8 | 4.515e+11 | 3.7 (0.97×) | 2.947e+11 (0.653×) |
| 64 | 64 | 5e-4 | 4.2 | 9.030e+11 | 4.2 (1.00× ) | 5.658e+11 (0.627×) |
| 64 | 64 | 5e-3 | 4.2 | 9.030e+11 | 4.3 (1.02×) | 5.304e+11 (0.587×) |
| 128 | 64 | 5e-4 | 4.4 | 9.047e+11 | 4.2 (0.96×) | 5.700e+11 (0.630×) |
| 128 | 64 | 5e-3 | 4.3 | 9.047e+11 | 4.2 (0.98×) | 5.406e+11 (0.598×) |
| 128 | 128 | 5e-4 | 4.8 | 1.809e+12 | 4.9 (1.02×) | 1.044e+12 (0.577×) |
| 128 | 128 | 5e-3 | 4.8 | 1.809e+12 | 5.1 (1.06×) | 1.000e+12 (0.553×) |
| 256 | 128 | 5e-4 | 4.2 | 1.817e+12 | 4.4 (1.05×) | 1.042e+12 (0.573×) |
| 256 | 128 | 5e-3 | 4.3 | 1.817e+12 | 4.2 (0.98×) | 1.007e+12 (0.554×) |
| 256 | 256 | 5e-4 | 4.4 | 3.634e+12 | 4.5 (1.02×) | 1.969e+12 (0.542×) |
| 256 | 256 | 5e-3 | 4.4 | 3.634e+12 | 4.7 (1.07×) | 1.920e+12 (0.528×) |
| 512 | 256 | 5e-4 | 4.0 | 3.667e+12 | 4.3 (1.08×) | 1.971e+12 (0.537×) |
| 512 | 256 | 5e-3 | 4.0 | 3.667e+12 | 4.2 (1.05×) | 1.934e+12 (0.528×) |
| 512 | 512 | 5e-4 | 4.5 | 7.334e+12 | 4.7 (1.04×) | 3.819e+12 (0.521×) |
| 512 | 512 | 5e-3 | 4.4 | 7.334e+12 | 4.8 (1.09×) | 3.777e+12 (0.515×) |

Table 3: Quality changes in generated responses by LLaDA-Instruct. Score$_{base}$ and Score$_{prop}$ indicate the overall score for the MT-Bench with single-turn settings evaluated with gpt-4o.

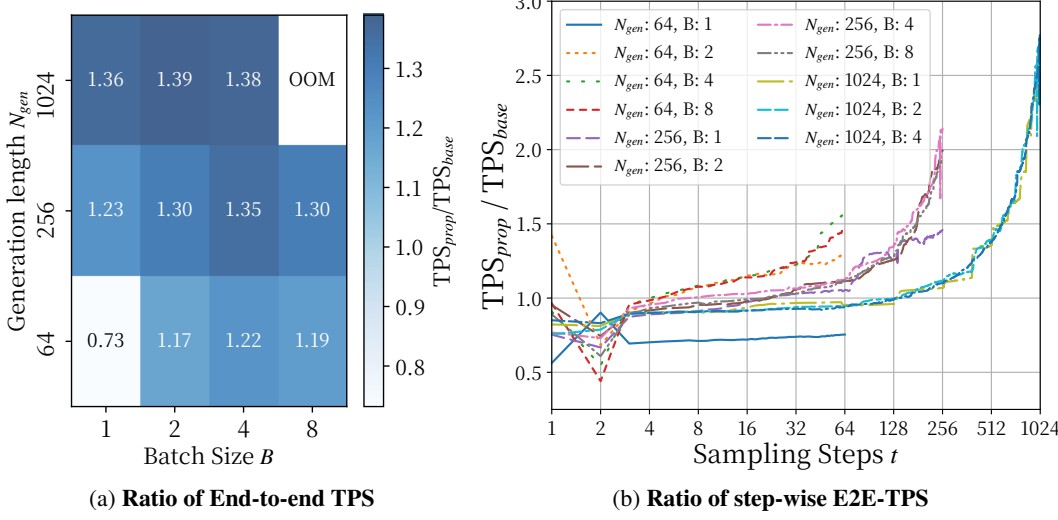

(a) **Ratio of End-to-end TPS**

(b) **Ratio of step-wise E2E-TPS**

Figure 3: Throughput behavior with SURELOCK: (a) end-to-end TPS ratio across different $N_{\text{gen}}$ and batch size $B$.; (b) per-step TPS ratio increasing as sampling progresses.

the TPS gains we report from our vanilla PyTorch implementation are conservative; Appendix L analyzes the cause of overheads in detail and discusses the possibility of hardware-specific optimizations (e.g., fused kernels) that could further align FLOP reductions with wall-clock speedups. See Figure 3b for temporal dynamics of step-wise TPS. We can see that the gain in local runtime increases as the step progresses, which is consistent with the computational trends (Figure 2).

**Experiment-5: Orthogonality to Existing Approaches for FLOPs/Runtime Improvement.** As already discussed in Section 1, Temporal-, Reuse-, and Selection-based approaches have optimization axes orthogonal to SureLock. While the number of queried positions remains proportional to N at every step, SureLock reduces the set of active positions that these operate on. Therefore, while we do not treat these acceleration methods as primary baselines, demonstrating that SureLock complements them in terms of FLOPs/Runtime improvements is still beneficial. For this purpose, we implement a surrogate version of selection-based method (Liu et al., 2025b) that computes only on a fixed fraction $k$ of active tokens at each step, and report algorithmic FLOPs/Runtime under three set-

| SureLock | Selection ($k = 0.8$) | $\uparrow$ TPS$_{prop}$/TPS$_{base}$ | $\downarrow$ $\mathcal{F}_{prop}$/$\mathcal{F}_{base}$ |
|:---:|:---:|:---:|:---:|
| ✓ | - | 1.30× | 0.54× |
| - | ✓ | 1.18× | 0,64× |
| ✓ | ✓ | **1.73×** | **0.28×** |

Table 4: Algorithmic FLOPs/Runtime for different acceleration methods, evaluated using LLaDA-8B-Instruct on MT-Bench with $N_{\text{gen}} = S = 256$.

| $N_{\text{gen}}$ | steps | $\varepsilon$ | $\downarrow$ Gen.-PPL$_{base}$ | $\downarrow$ $\mathcal{F}_{base}$ | $\downarrow$ Gen.-PPL$_{prop}$ | $\downarrow$ $\mathcal{F}_{prop}$ |
|:---:|:---:|:---:|:---:|:---:|:---:|:---:|
| 64 | 64 | 5e-8 | 3.4813 | 8.976e+11 | 4.0033 (1.14×) | 5.930e+11 (0.66×) |
| 64 | 64 | 5e-7 | 3.4813 | 8.976e+11 | 4.3489 (1.25×) | 5.430e+11 (0.60×) |
| 64 | 64 | 5e-4 | 3.4813 | 8.976e+11 | 4.5722 (1.31×) | 5.203e+11 (0.58×) |
| 64 | 64 | 5e-3 | 3.4813 | 8.976e+11 | 4.7596 (1.37×) | 4.664e+11 (0.52×) |

Table 5: Generation quality of continuation sequences with and without SURELOCK on LLaDA-8B-Base on WikiText-103 for different values of $\varepsilon$.

tings: i) selection-based ($k = 0.8$), ii) SureLock, and iii) their combination (i,e., selecting the fixed fraction $k = 0.8$ of positions among positions not yet locked). Note that, for a focused comparison, we did not search over $k$ for optimal acceleration, so the results of the selection baseline should be interpreted as conservative; our goal here is to illustrate the relative behavior. Table 4 shows that the combination yields additional acceleration over either component alone, supporting our claim that SureLock and selection-based sparsification cooperate rather than compete.

**Experiment-6: Room for Optimization of Locking Threshold.** In our main experiments (Table 2 and Table 3), we deliberately used a fixed set of global locking thresholds across all configurations, rather than tuning it per setting, aiming to reveal the intrinsic behavior of SureLock under a wide range of generation length. This choice makes it clear that the method is more effective in compute-heavy settings (longer outputs and more reverse steps), while the configuration is relatively aggressive for short generations. However, as in Theorem 1, by tightening $\varepsilon$, we can reduce the final distributional error. Here, we demonstrate the potential for $\varepsilon$-optimization by sweeping $\varepsilon$. We adopt the settings of short sequence language modeling using LLaDA-8B-Base here, which exhibited relatively lower quality in the main experiments. Table 5 shows that, even for short generations, an optimal $\varepsilon$ exists that minimizes quality degradation while still providing nontrivial acceleration.

**Example Analysis.** Feeding the same question sampled from MT-Bench to the LLaDA-8B-Instruct, Figure 4 shows the responses from the same sampler with and without SURELOCK. With settings achieving approximately $0.6\times$ of computational complexity on average (See Table 3), we can confirm that not only the quantitative score but also the qualitative quality is comparable to the baseline, with minor, localized variations. While locking might seem to introduce disrupted generation, the impact is limited. Furthermore, we confirmed in code generation that these minor differences do not disrupt the sentence syntax (Appendix J). More examples are in Appendix N.

## 4 RELATED WORK

**Computational savings for DLMs.** Prior efforts to decrease the sampling cost of discrete diffusion LMs (DLMs) largely follow two axes—*Temporal* and *Reuse*. *Temporal* methods shrink the number of sampling steps $T$ via parallel/dilated unmasking and edit-based updates (Ghazvininejad et al., 2019; Gu et al., 2019; Luxembourg et al., 2025; Israel et al., 2025; Huang & Tang, 2025; Wei et al., 2025), or by adopting higher-order / distilled samplers from diffusion literature, e.g., DPM-Solver (Lu et al., 2022). *Reuse* methods amortize work across steps by reusing or approximating intermediate states—e.g., $K/V$ caching or approximation and selective feature refresh (Ma et al., 2025; Liu et al., 2025b; Wu et al., 2025a), Selection-based partial compute reduces per-step complexity by updating only a fixed number of subset positions, sometimes with periodic full recomputation for stability (Liu et al., 2025b; Wu et al., 2025a). These two axes primarily reduce *how many steps* we take or *how much previous computation* we can reuse between steps; they typically leave

> **Question:** Benjamin went to a bookstore and purchased a variety of books. He bought 5 copies of a sci-fi novel, each priced at $20, 3 copies of a history book priced at $30 each, and 2 copies of a philosophy book for $45 each. What was the total cost of his purchases?.

**Baseline** (score: 10)
Benjamin bought 5 copies of a sci-fi novel, each priced at $20, so he spent 5 * $20 = $100. He bought 3 copies of a history book, each priced at $30, so he spent 3 * $30 = $90. He bought 2 copies of a philosophy book, each priced at $45, so he spent 2 * $45 = $90. The total cost of his purchases is $100 + $90 + $90 = $280. Summing up,

**SureLock** (score: 10)
Benjamin bought 5 copies of a sci-fi novel for $20 each, so the total cost is 5 * $20 = $100. He also bought 3 copies of a history book for $30 each, so the total cost is 3 * $30 = $90. Lastly, he bought 2 copies of a philosophy book for $45 each, so the total cost is 2 * $45 = $90. The total cost of his purchases is $100 + $90 + $90 = $280. Con

Figure 4: Comparison of responses between Baseline vs. SURELOCK on LLaDA-8B-Instrut with $\varepsilon = 5e - 4$, $N_{\text{gen}} = 128$, $S = 128$. The question is sampled from MT-bench with question id= 119.

the *within-step* active-query count constant up to the end of the sampling step, so per-step compute remains bounded by $N$ even late in sampling. Crucially, these are complementary to SURELOCK; by focusing only on the gradually contracting active positions, they could yield even greater savings.

**DLMs on longer sequences.** Scaling DLMs to longer contexts has been actively explored such as span/block-wise masking (Arriola et al., 2025), adaptive masked token insertion (Arriola et al., 2025), DLMs' suite Rotary Position Embedding (RoPE) Liu et al. (2025a). This line of research directions yields diverse applications such as reasoning on massive code bases. By contrast, decoding longer sequences by DLMs generally requires more sampling steps, making the stationary per-step computational cost more severe. Advancing techniques such as SURELOCK, which monotonically reduces this stationary compute as sampling proceeds, could facilitate DLMs' research on longer contexts, which lags behind compared to AR models (Wu et al., 2025b), i.e., thousands vs. millions.

## 5 CONCLUSION

We introduced SURELOCK, a method for iterative decoding in masked-diffusion LMs (MDLMs) that locks converged token positions and thereafter bypasses their query projection and per-token FFN, yielding a *monotonic* reduction in per-step compute. Locked positions remain fully attendable via cached $K/V$ vectors. The core design is a *local KL* locking test; we justified this approach by deriving a closed-form link between the lock-time local KL and an error bound on the terminal log-probability. On language modeling and instruction-following tasks, SURELOCK achieves large reductions in algorithmic FLOPs while maintaining task quality.

**Future work.** Our approach is orthogonal to step-count reduction and inter-step reuse techniques, as well as selection approaches (Sec. 1); such methods can operate on the progressively shrinking active positions induced by SURELOCK. Experiment 5 shows that combining with an orthogonal approach improves acceleration; full co-optimization is left for future work. Our "converge-then-lock" idea is modality-agnostic, but our KL-based formulation assumes discrete posteriors; a continuous version would require explicit local uncertainty models and a new theoretical treatment.

**Limitations.** We report efficiency in terms of algorithmic FLOPs, a kernel- and hardware-agnostic proxy for theoretical savings that decouples the sampling algorithm from wall-clock implementations. Theorem 1 is used only to justify the KL-based locking rule rather than to produce numeric thresholds; such model- and data-specific tuning is left for future work. Our locking rule is based on step-wise distributional stability in the posterior (Sec. 2.2), which does not formally guarantee that semantics are fully determined, while we observe no serious degradation in our benchmarks probably because SURELOCK only locks confidently unmasked positions (Appendix E).

## ETHICS STATEMENT

This paper aims to reduce computational complexity in inference time for discrete diffusion models, which may have the potential to significantly impact real-world applications. Furthermore, this research does not require diffusion models to generate harmful or sexual content. We believe our work does not directly contribute to malicious use for such purposes.

## REPRODUCIBILITY STATEMENT

Experimental settings are included in Sec. 3. More detailed descriptions are provided in the Appendices. We provide more details at the project page: https://daioba.github.io/surelock.

## ACKNOWLEDGEMENT

This work was partially supported by JSPS KAKENHI Grant Number 25H01137 and JST K Program Japan Grant Number JPMJKP24C3.

Danushka Bollegala holds concurrent appointments as a Professor at University of Liverpool and as an Amazon Scholar. This paper describes work performed at the University of Liverpool and is not associated with Amazon.

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

## A    APPENDIX

## B    USAGE OF LARGE LANGUAGE MODELS

We used LLMs to assist with writing paragraphs and searching literature. However, we retain full responsibility for the content in this paper.

## C    EMPIRICAL INVESTIGATION OF ASSUMPTION A2

The assumption of monotonic decrease of the step-wise KL divergence in the reverse step in Theorem 1 is, in principle, theoretical. However, it is informative to gain a better understanding of how it behaves in practice; we visualize the dynamics of the step-wise KL divergence in Figure 5. We used LLaDA-8B-Instruct with $\varepsilon = 5e - 4$, $N_{\text{gen}} = 128$, and $S = 128$ , on MT-Bench's 16 samples. We can see that the Step-wise KL indeed monotonically decreases as the sampling proceeds; we can infer that A2 is not merely a theoretical assumption, and it does not significantly deviate from the practical situation.

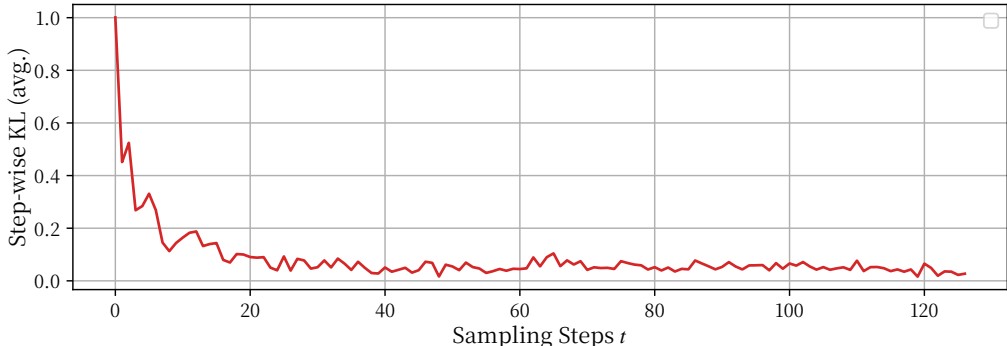

Figure 5: **Dynamics of Step-wise KL divegence.** Averaged step-wise KL divergence measured using LLaDA-8B-Instruct on MT-Bench prompts.

## D    DETAILED EXPLANATION OF ASSUMPTION A3.

We show that the one–step deviation of the logit vector $z_s^{(i)}$ at position $i$ is controlled by the *previous* step's drift on local posterior measured by KL divergence. Throughout, let $p_t^{(j)} \in \Delta^{V-1}$ be the token posterior at position $i$ and step $t$, $D_t^{(j)} \coloneqq \mathrm{KL}\big(p_t^{(j)} \,\|\, p_{t-1}^{(j)}\big)$, and let $E \in \mathbb{R}^{V \times d_{\text{model}}}$ denote the embedding matrix (rows are token embeddings). We use the block norm $\|\cdot\|_{2,1}$ defined by $\|X\|_{2,1} = \sum_j \|x_j\|_2$ for $X = [x_1, \ldots, x_n]^\top$.

**Reclaiming Assumption A3.**    There exists a constant $L > 0$, depending only on operator norms of the network weights and softmax/LN/activation Lipschitz constants, such that for all steps $s$ and positions $i$,

$$\|z_s^{(i)} - z_{s-1}^{(i)}\|_2 \ \leq \ L \sqrt{D_{s-1}^{(i)}}. \tag{1}$$

Below we prove a slightly stronger *global* bound depending on $\sum_j \sqrt{D_{s-1}^{(j)}}$ and then specialize it to the local form equation 1 using a locality factor.

### D.1    FROM POSTERIOR DRIFT TO EXPECTED-EMBEDDING DRIFT

For any $q, r \in \Delta^{V-1}$ define the expected input (embedding) $x(q) \coloneqq E^\top q \in \mathbb{R}^{d_{\text{model}}}$. Then

$$\|x(q) - x(r)\|_2 = \|E^\top(q - r)\|_2 \le \|E\|_2 \|q - r\|_2 \le \|E\|_2 \|q - r\|_1. \tag{A.3.1}$$

By Pinsker, $\|q - r\|_1 \leq \sqrt{2\,\mathrm{KL}(q\|r)}$, so with $L_{\mathrm{emb}} := \|E\|_2 \sqrt{2}$ we have

$$\|x(q) - x(r)\|_2 \ \leq \ L_{\mathrm{emb}} \ \sqrt{\mathrm{KL}(q\|r)}. \tag{A.3.2}$$

Applying equation A.3.2 to successive steps at position $i$,

$$\Delta x_s^{(j)} \ := \ x\big(p_{s-1}^{(j)}\big) - x\big(p_{s-2}^{(j)}\big), \qquad \|\Delta x_s^{(j)}\|_2 \ \leq \ L_{\mathrm{emb}} \sqrt{D_{s-1}^{(j)}}. \tag{A.3.3}$$

## D.2 SINGLE-HEAD ATTENTION: LIPSCHITZ BOUND

Consider a single-head attention with parameters $(W_Q, W_K, W_V)$ and key/query dimension $d_k$. For input $X = [x_1, \ldots, x_n]^\top$, the head output at position $i$ is

$$o_i(X) = \sum_{j=1}^n \alpha_{ij}(X)\, W_V x_j, \qquad \alpha_{i\bullet}(X) = \mathrm{softmax}\Big(\tfrac{1}{\sqrt{d_k}}(W_Q x_i)(W_K X)^\top\Big). \tag{A.3.4}$$

For two inputs $X$ and $X' = X - \Delta X$, a standard three-term decomposition gives

$$\Delta o_i \ := \ o_i(X) - o_i(X') \ = \ \underbrace{\sum_j \alpha'_{ij}\, W_V\, \Delta x_j}_{(I)} + \underbrace{\sum_j (\alpha_{ij} - \alpha'_{ij})\, W_V\, x'_j}_{(II)} + \underbrace{\sum_j (\alpha_{ij} - \alpha'_{ij})\, W_V\, \Delta x_j}_{(III)}. \tag{A.3.5}$$

We upper bound $(I)$ and $(II)$ and subsume $(III)$ into them (yielding a conservative bound).

**Term (I)** By the triangle inequality,

$$\|(I)\|_2 \leq \|W_V\|_2 \sum_j \alpha'_{ij}\, \|\Delta x_j\|_2 \leq \|W_V\|_2 \sum_j \|\Delta x_j\|_2 = \|W_V\|_2\, \|\Delta X\|_{2,1}. \tag{A.3.6}$$

**Softmax Jacobian bound.** Let $\alpha = \mathrm{softmax}(s)$ and $J_{\mathrm{sm}}(s) = \mathrm{diag}(\alpha) - \alpha\alpha^\top$. It is classical that

$$\sup_s \|J_{\mathrm{sm}}(s)\|_2 = \tfrac{1}{2}, \tag{A.3.7}$$

achieved at two-point distributions. Hence, with mean value theorem, for score vectors $s_i$ and $s'_i$,

$$\|\alpha_{i\bullet} - \alpha'_{i\bullet}\|_2 \leq \tfrac{1}{2}\, \|s_i - s'_i\|_2, \qquad \|\alpha_{i\bullet} - \alpha'_{i\bullet}\|_1 \leq \sqrt{n}\, \|\alpha_{i\bullet} - \alpha'_{i\bullet}\|_2. \tag{A.3.8}$$

**Score difference.** Write the $(i,j)$ score as

$$s_{ij} = \tfrac{1}{\sqrt{d_k}} \langle W_Q x_i,\, W_K x_j \rangle. \tag{A.3.9}$$

Then

$$|\Delta s_{ij}| = \tfrac{1}{\sqrt{d_k}} \big| \langle W_Q \Delta x_i,\, W_K x'_j \rangle + \langle W_Q x_i,\, W_K \Delta x_j \rangle + \langle W_Q \Delta x_i,\, W_K \Delta x_j \rangle \big|$$
$$\leq \tfrac{\|W_Q\|_2 \|W_K\|_2}{\sqrt{d_k}} \big( \|\Delta x_i\|_2\, \|x'_j\|_2 + \|x_i\|_2\, \|\Delta x_j\|_2 + \|\Delta x_i\|_2\, \|\Delta x_j\|_2 \big). \tag{A.3.10}$$

Assume a uniform radius bound $\|x_j\|_2, \|x'_j\|_2 \leq R_x$ for all $j$ (true if inputs are bounded and LN is used). Summing equation A.3.10 over $j$ and using Cauchy–Schwarz and $\|\Delta X\|_{2,1} = \sum_j \|\Delta x_j\|_2$ yields

$$\|\Delta s_{ij}\|_2 = \sqrt{\sum_{j=1}^n \Delta(s_{i,j})^2} \leq \tfrac{\|W_Q\|_2 \|W_K\|_2}{\sqrt{d_k}} \big( \sqrt{n}\, R_x\, \|\Delta x_i\|_2 + R_x\, \|\Delta X\|_{2,1} + \|\Delta x_i\|_2\, \|\Delta X\|_{2,1} \big). \tag{A.3.11}$$

**Term (II).** Using the triangle inequality and operator norm of $W_V$, we have

$$\|(II)\|_2 = \left\| \sum_j (\alpha_{ij} - \alpha'_{ij}) W_V x'_j \right\|_2 \leq \|W_V\|_2 \sum_j |\alpha_{ij} - \alpha'_{ij}| \, \|x'_j\|_2. \tag{A.3.12a}$$

Assuming $\|x'_j\|_2 \leq R_x$ for all $j$, we obtain

$$\|(II)\|_2 \leq \|W_V\|_2 R_x \|\alpha_{i\bullet} - \alpha'_{i\bullet}\|_1. \tag{A.3.12b}$$

Using the norm inequality $\| \cdot \|_1 \leq \sqrt{n} \, \| \cdot \|_2$,

$$\|(II)\|_2 \leq \|W_V\|_2 R_x \sqrt{n} \, \|\alpha_{i\bullet} - \alpha'_{i\bullet}\|_2. \tag{A.3.12c}$$

Finally, applying the softmax Jacobian bound $\|\alpha_{i\bullet} - \alpha'_{i\bullet}\|_2 \leq \frac{1}{2} \|s_i - s'_i\|_2$ (cf. equation A.3.8), we arrive at

$$\|(II)\|_2 \leq \tfrac{\|W_V\|_2 \sqrt{n} R_x}{2} \|s_i - s'_i\|_2. \tag{A.3.12}$$

Substituting equation A.3.11 into equation A.3.12, we obtain

$$\|(II)\|_2 \leq \tfrac{\|W_V\|_2 \sqrt{n} R_x}{2} \cdot \tfrac{\|W_Q\|_2 \|W_K\|_2}{\sqrt{d_k}} \left( \sqrt{n} R_x \|\Delta x_i\|_2 + R_x \|\Delta X\|_{2,1} + \|\Delta x_i\|_2 \|\Delta X\|_{2,1} \right). \tag{A.3.13a}$$

Using $\|\Delta x_i\|_2 \leq \|\Delta X\|_{2,1}$, this simplifies to

$$\|(II)\|_2 \leq \|W_V\|_2 \cdot \tfrac{\|W_Q\|_2 \|W_K\|_2}{2\sqrt{d_k}} \cdot nR_x^2 \|\Delta X\|_{2,1}. \tag{A.3.13b}$$

Therefore, combining equation A.3.6 and equation A.3.13b, we obtain the single-head bound

$$\|\Delta o_i\|_2 \leq A_{\text{att}} \|\Delta X\|_{2,1}, \qquad A_{\text{att}} := \|W_V\|_2 \left[ 1 + \tfrac{\|W_Q\|_2 \|W_K\|_2}{2} \tfrac{n R_x^2}{\sqrt{d_k}} \right]. \tag{A.3.13}$$

### D.3 BLOCK COMPOSITION: MHA + FFN + RESIDUAL + LN

Let one Transformer block be

$$B(X) = X + \text{FFN}(\text{MHA}(\text{LN}(X))). \tag{A.3.14}$$

Assume Lipschitz bounds for each component (w.r.t. $\| \cdot \|_{2,1}$)

$$\|\text{LN}(X) - \text{LN}(X')\|_{2,1} \leq L_{\text{ln}} \|X - X'\|_{2,1}, \quad \|\text{MHA}(U) - \text{MHA}(U')\|_{2,1} \leq A_{\text{mha}} \|U - U'\|_{2,1}, \tag{A.3.15}$$

and

$$\|\text{FFN}(V) - \text{FFN}(V')\|_{2,1} \leq A_{\text{ff}} \|V - V'\|_{2,1} \tag{A.3.16}$$

For multi-head attention with $h$ heads concatenated and an output projection $W_O$, one may take

$$A_{\text{mha}} \leq \|W_O\|_2 \left( \max_{\text{head}} A_{\text{att}}^{(\text{head})} \right). \tag{A.3.17}$$

By the residual form equation A.3.14,

$$\|B(X) - B(X')\|_{2,1} \leq \|X - X'\|_{2,1} + A_{\text{ff}} A_{\text{mha}} L_{\text{ln}} \|X - X'\|_{2,1} = L_{\text{blk}} \|X - X'\|_{2,1}, \tag{A.3.18}$$

with

$$L_{\text{blk}} := 1 + A_{\text{ff}} A_{\text{mha}} L_{\text{ln}}. \tag{A.3.19}$$

For a stack of $H$ blocks,

$$\|B^H(X) - B^H(X')\|_{2,1} \leq L_{\text{blk}}^H \|X - X'\|_{2,1}. \tag{A.3.20}$$

Finally, with readout $z_i = W_o h_i$, we get

$$\|z_i(X) - z_i(X')\|_2 \leq \|W_o\|_2 \|B^H(X) - B^H(X')\|_{2,1} \leq \underbrace{\|W_o\|_2 L_{\text{blk}}^H}_{L_{\text{net}}} \|X - X'\|_{2,1}. \tag{A.3.21}$$

### D.4 PUTTING THINGS TOGETHER

Applying equation A.3.21 to successive sampler states $(X_s, X_{s-1})$ and using equation A.3.3,

$$\|z_s^{(i)} - z_{s-1}^{(i)}\|_2 \leq L_{\text{net}} \|X_s - X_{s-1}\|_{2,1} = L_{\text{net}} \sum_j \|\Delta x_s^{(j)}\|_2 \leq L_{\text{net}} L_{\text{emb}} \sum_j \sqrt{D_{s-1}^{(j)}}.$$

(A.3.22)

Define $L_{\text{all}} := L_{\text{net}} L_{\text{emb}}$. Equation A.3.22 is the *global* for if equation 1.

$$\|z_s^{(i)} - z_{s-1}^{(i)}\|_2 \ \leq \ L_{\text{all}} \sum_j \sqrt{D_{s-1}^{(j)}}.$$

(A.3.23)

Late in iterative sampling the re-masking rate decreases and many positions become stable. Empirically one can upper bound the tail contribution at step $s-1$ by

$$\sum_{j \neq i} \sqrt{D_{s-1}^{(j)}} \ \leq \ \kappa_{s-1}^{(i)} \sqrt{D_{s-1}^{(i)}}, \qquad \kappa_{s-1}^{(i)} \in [0, \infty),$$

(A.3.24)

where $\kappa_{s-1}^{(i)}$ is an observable ratio. Substituting equation A.3.24 into equation A.3.23,

$$\|z_s^{(i)} - z_{s-1}^{(i)}\|_2 \ \leq \ L_{\text{all}} \left(1 + \kappa_{s-1}^{(i)}\right) \sqrt{D_{s-1}^{(i)}} \ = \ \underbrace{L_{\text{all}}(1 + \kappa_{s-1}^{(i)})}_{=: L} \sqrt{D_{s-1}^{(i)}}.$$

(A.3.25)

This is exactly Assumption A3 in the main text.

**Constants at a glance.**

$$L \ = \ \underbrace{\|E\|_2 \sqrt{2}}_{L_{\text{emb}}} \times \underbrace{\|W_o\|_2 \left(1 + A_{\text{ff}} A_{\text{mha}} L_{\text{ln}}\right)^H}_{L_{\text{net}}} \times (1 + \kappa_{s-1}^{(i)}), \quad A_{\text{mha}} \leq \|W_O\|_2 \max_{\text{head}} A_{\text{att}}^{(\text{head})},$$

with $A_{\text{att}}^{(\text{head})}$ given by equation A.3.13 and $A_{\text{ff}} = \|W_2\|_2 L_{\text{act}} \|W_1\|_2$.

## E SEMANTIC STABILITY ON KL-BASED LOCKING

Our locking rule is driven by distributional stability; we lock a token position when the KL divergence between successive posteriors at that position becomes small and remains small across steps. Conceptually, however, low KL between successive posteriors does not strictly guarantee that the semantics at that position are fully determined. In principle, one can imagine a position whose distribution remains almost unchanged across steps while probability mass oscillates between two near-synonymous tokens (e.g., "happy" and "joyful"), which would still yield a small step-wise KL.

We regard such cases as essentially benign. Choosing one of several near-synonymous options has little impact on the overall semantic content of the sentence; in this scenario, locking effectively commits to one paraphrase among a small set of semantically equivalent alternatives.

More concerning would be different cases when the model is still undecided between *semantically distant* options, such as "he" vs. "she", "yes" vs. "no", or "positive" vs. "negative". In that case, prematurely locking could have a much larger impact on the downstream continuation.

In practice, however, the combination of the masked diffusion sampler and our locking policy makes this latter situation rare. First, standard masked diffusion schedulers unmask [MASK] positions in order of confidence: at each step, they select a fixed (or adaptive) number of positions with the highest posterior probability and convert them from [MASK] to concrete tokens. Positions where the model is genuinely uncertain between semantically distant options typically have relatively flat posteriors and are therefore unmasked in later steps. Second, SureLock only applies the KL-based locking rule to positions that have already been unmasked. Thus, by the time a position is both unmasked and classified as "stable" by our criterion, the model usually assigns high probability to a narrow set of semantically consistent tokens, and any residual uncertainty is mostly between close paraphrases rather than contradictory alternatives.

Empirically, this intuition is supported by our benchmarks. Across a wide range of configurations in Tables 2, 3, and 6 (varying tasks, generation length, number of diffusion steps, and other hyperparameters), enabling locking does not lead to systematic or severe degradation in either language modeling metrics or instruction-following performance relative to the unlocked baseline. That means the differences we do observe are typically localized, i.e., slightly different word choices or phrasing around a locked position rather than global breakdowns of topic, discourse structure, or adherence to the input instruction. Appendix N provides side-by-side examples of baseline vs. SureLock outputs under our default settings.

## F  INTERACTION BETWEEN SAMPLING TEMPERATURE AND SURELOCK

In our experiments, we fixed the sampling temperature $\tau$ to 0 to isolate the effect of SureLock. Importantly, however, our locking mechanism itself does not rely on a particular sampling temperature.

The locking criterion (Sec. 2.2) is computed on the raw posterior before temperature scaling, i.e., using the distribution obtained from the raw logits (equivalently, a $\tau = 1$ softmax). The sampling temperature $\tau$, when non-zero, is applied only in the categorical sampling step that materializes discrete tokens, not in the distributions used for the KL-based stability test. Consequently, for a fixed threshold $\varepsilon$, the set of positions judged "stable" by SureLock is independent of $\tau$: increasing $\tau$ changes the diversity of sampled outputs, but does not by itself delay or accelerate locking timing.

One could alternatively apply the same temperature transform to the posteriors before computing KL, i.e., base the stability test on $p_t^\tau$ and $p_{t-1}^\tau$ rather than $p_t$ and $p_{t-1}$. Mathematically, this defines a temperature-dependent divergence that changes which parts of the distribution differences are emphasized, e.g., flattening posteriors makes KL smaller and relaxes the notion of "stability".

In practice, however, this mainly reparameterizes the sensitivity of the locking test and plays a quite similar role to adjusting the threshold $\varepsilon$ itself. Therefore, for clarity and simplicity, we keep the locking criteria temperature-agnostic and let $\varepsilon$ control how tolerant SureLock is to distributional changes, while temperature controls decoding diversity.

## G  DATASET SETTINGS FOR EXPERIMENT-1

To efficiently sweep a large configuration space, in experiment-1 to profile algorithm FLOPs, we used a focused subset of MT-Bench rather than the full set. For each of the eight categories e.g., *"writing"* and *"coding,"* we use the first four single-turn prompts—32 prompts in total—so that diverse phenomena may appear. Prompts are used unedited and are preprocessed with the LLADA-8B-Instruct tokenizer using its default chat template; no system message is injected.

## H  ADDITIONAL RESULTS FOR EXPERIMENT-2

We show additional results for different settings of $m$-percentile used for the optional confidence gate 2.2; we set $m = \{10\%, 40\%\}$, which differ from the one in the main text i.e., $m = 20\%$. The trend in temporal dynamics of FLOPs ratio does not change significantly with $m$, suggesting $m$ is not a particularly sensitive parameter.

## I  DETAILED SETTINGS FOR EXPERIMENT-3

### I.1  ON WIKITEXT-103

We extracted a small subset from Wikitext-103 (Merity et al., 2016), which contains diverse text, for the continuation generation task. First, we removed blank lines from the dataset loaded from `https://huggingface.co/datasets/Salesforce/wikitext`, then extracted the first 120 records at odd indices. Next, we extracted the first 64 tokens from each record to form the prompt $x$ for the continuation. We here used LLaDA-8B-Base tokenizer (Nie et al., 2025).

For each prompt $x$, we generate a sequence $y$ of fixed length $N_g$, then compute Gen.-PPL using an external autoregressive language model $\theta$. This work uses Llama-3-8B (Grattafiori et al.,

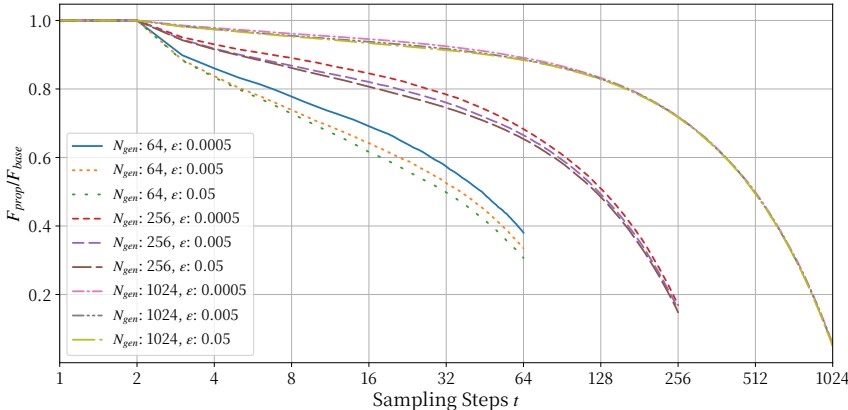

Figure 6: **Step-wise FLOPs ratio.** Algorithmic FLOPs ratio at step $t$ i.e., $\mathcal{F}_{\text{prop}}^t/\mathcal{F}_{\text{base}}^t$ (and the averaged per-step active row ratio (micro) $\bar{r}_t = \sum_b M_{t,b}/\sum_b B\,N_b$ consistently decreases as steps proceed, explaining later-step savings of computational cost. $m$-percentile is $10\%$.

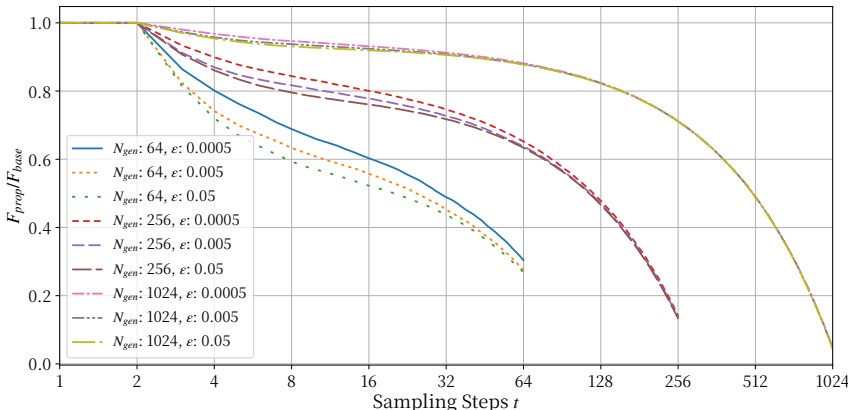

Figure 7: **Step-wise FLOPs ratio.** Algorithmic FLOPs ratio at step $t$ i.e., $\mathcal{F}_{\text{prop}}^t/\mathcal{F}_{\text{base}}^t$ (and the averaged per-step active row ratio (micro) $\bar{r}_t = \sum_b M_{t,b}/\sum_b B\,N_b$ consistently decreases as steps proceed, explaining later-step savings of computational cost. $m$-percentile is $40\%$.

2024), a representative autoregressive language model. We define Gen.-PPL as perplexity micro-averaged over the dataset; we first compute the micro-averaged negative log-likelihood $\overline{\text{NLL}}_{\text{micro}} = \sum_i \text{NLL}_i / \sum_i |y_i|$, then $\text{PPL}_{\text{micro}} = \exp(\overline{\text{NLL}}_{\text{micro}})$. In the report, we used natural logarithms.

## I.2 On MT-Bench

We evaluated LLaDA-8B-Instruct (Nie et al., 2025) using all 80 records in MT-bench Zheng et al. (2023) with a single-turn configuration—we used the first prompt for each record. Essentially, using the LLaDA-8B-Instruct tokenizer, we applied its default chat template to each prompt to form input. We then fed these prompts to the model to generate a responses.

Response evaluation was conducted using the LLM-as-a-judge format. Specifically, we used the official MT-Bench evaluation repository https://github.com/lm-sys/FastChat/tree/main/fastchat/llm_judge as-is. As the judge model, we employed OpenAI's gpt-4o.

## J Challenging Set for Experiment-3

We conduct additional experiments for Experiment-3 as a challenging set. We use code generation, where it is examined by running generated code in a sandbox environment to report pass rate

(Pass@1). Therefore, it makes it a stringent test of whether premature locking induces harmful failures. We evaluate HumanEval's first 70 examples (Chen et al., 2021) under a representative configuration, i.e., LLaDA-8B-Instruct, with $N_{\text{gen}} = S = 256$, as used in Experiment-3. Under the setting, the baseline achieves a pass@1 of 18.6, while SureLock achieves 18.6-20.3 with about 0.5× efficiency in algorithmic FLOPs. We can see that SureLock operates even in code generation benchmarks where finite surface fluctuations directly impact performance, reducing computational cost while maintaining original performance.

| $N_{\text{gen}}$ | steps | $\varepsilon$ | $\uparrow \text{Pass@1}_{base}$ | $\downarrow \mathcal{F}_{base}$ | $\uparrow \text{Pass@1}_{prop}$ | $\downarrow \mathcal{F}_{prop}$ |
|---|---|---|---|---|---|---|
| 256 | 256 | 5e-4 | 18.6 | 3.648e+12 | 20.3 (1.09×) | 1.716e+12 (0.470×) |
| 256 | 256 | 5e-3 | 18.6 | 3.648e+12 | 18.6 (1.00×) | 1.676e+12 (0.459×) |

Table 6: Quality changes in generated responses by LLaDA-Instruct. $\text{Pass@1}_{base}$ and $\text{Pass@1}_{prop}$ indicate the pass rate (%) of the test cases for HumanEval.

## K    RUNTIME METRICS

*E2E-TPS* measures the sustained decoding throughput of the model across multiple batches, including all inter-batch gaps and host/device launch overheads incurred during decoding. Concretely, after preparing batches we synchronize the device and start a single wall-clock timer immediately before the first decoding call, and stop it after the last decoding finishes. The numerator is the total number of unmasked tokens produced by the iterative sampling process.

$$\text{E2E-TPS} = \frac{\#\{\text{total unmasked tokens}\}}{\text{wall-clock time}}$$

*Step TPS* at sampling step $t$ is computed as the number of newly unmasked tokens at that step divided by the wall-clock time, aggregating across batches:

$$\text{StepTPS}(t) = \frac{\#\{\text{newly unmasked tokens at step } t\}}{\Delta t}$$

where $\Delta t$ is the wall-clock time for that step $t$. In both baseline and SURELOCK, timing uses CUDA events with synchronization.

## L    ALGORITHMIC CHARACTERISTICS ON RUNTIME

While our method yields substantial reductions in algorithmic FLOPs, the corresponding wall-clock speedups are more modest in some settings. This gap arises from several implementation-level overheads.

Reusing locked tokens requires irregular reads and writes to the computed kernels (e.g., Q/K/V) and caches using non-contiguous indices (Algorithm 1; Line10-13), causing scattered, poorly coalesced memory accesses and making the system memory-bound. Even when many tokens are skipped, a large amount of K/V values must still be transferred between the cache and the compute kernels (Algorithm 1; Line 11), so cache traffic can dominate latency in non-compute-bound regimes. In addition, each diffusion step incurs a fixed cost for managing active and frozen tokens, such as computing posterior KL values, applying thresholds, constructing index sets, and updating the cache (Algorithm 1, Lines 17-25), which becomes non-negligible once the per-step arithmetic workload is small. Finally, when only a small subset of tokens remains active, generic dense attention and FFN kernels operate on thin slices of the sequence dimension (Algorithm 1, Lines 10 and 12), resulting in low thread/warp occupancy and underutilization of the hardware.

Our current implementations are based on standard PyTorch operators and off-the-shelf dense kernels on commodity GPUs, without the use of custom kernels, kernel fusion, or specialized cache layouts. Consequently, the reported wall-clock improvements should be viewed as conservative.

In the future, we believe it will be possible to further bridge the gap between computational complexity reduction and real-time speed improvements by introducing hardware-specific optimizations that are orthogonal to our algorithmic contribution.

In particular, one could exploit (i) cache layout redesign and periodic compaction so that active tokens are packed into (block-)contiguous regions to restore coalesced loads and stores from the K/V caches; (ii) dedicated fused kernels that, given the current active position set, jointly perform index gathering, attention and FFN computation, and writeback into the compact cache within a single launch, thereby reducing kernel-launch overhead and intermediate global-memory traffic; and (iii) overlap of cache operations with compute by issuing asynchronous copies and prefetches of cached K/V segments for the next step while computing attentions and FFN for the current step, effectively hiding much of the cache-access latency.

## M  Optional Unlocking

In SURELOCK, once locked, a token position is *excluded from re-masking by construction* (Sec. 2). However, rare context shifts can render a cached posterior stale. Therefore, allowing lightweight *unlocking* to detect context drift within a small finite budget and returning tokens to the active set could also be considered as a useful option. In the following, we discuss an implementation for that option.

After every $P$ steps, we evaluate a budgeted proxy on locked rows only (e.g., variable-length attention with a reduced subgraph). This yields proxy uncertainty $\tilde{u}_t^{(i)}$ and distributional drift from the locked time $\widetilde{D}_t^{(i)} = \mathrm{KL}(\hat{p}_t^{(i)} \| p_{t_i^\star}^{(i)})$. Let $\theta_t = q_m(u_t)$ be the top-$m\%$ uncertainty among *active* tokens at step $t$. We *unlock* a locked token $i$ if any holds:

$$\tilde{u}_t^{(i)} > \theta_t \quad \wedge \quad \widetilde{D}_t^{(i)} > \varepsilon_{\text{unlock}} \quad \wedge \quad t - t_i^\star > D_{\text{interval}}. \tag{2}$$

$D_{\text{interval}}$ is set for suppressing failure to converge due to excessive oscillation in the locking and unlocking behavior. Furthermore, after an unlock, position $i$ cannot be re-locked for $D_{\text{cool}}$ steps and must satisfy a stricter KL threshold $\rho\varepsilon$ ($\rho \leq 1$) when re-locking.

On unlock at step $t$, we invalidate cached $K/V$ for $i$ and reintroduce its Query projection and FFN sublayers from step $t+1$; the current step is unchanged. Amortized probe overhead on attention-dominant FLOPs is a trade-off with performance. However, the subgraph used for probing can be reduced to a size negligible compared to the dominant cost, depending on its size.

Let $\Delta_t^{(i)} = \| \log p_t^{(i)} - \log p_t^{(i),\text{full}} \|_2$ be deviation from a no-lock decode. Unlocking cannot increase $\Delta_{t'}^{(i)}$ for $t' \geq t$ because the row is computed by the same subgraph as baseline thereafter. Therefore, the error bound (Theorem 1 in Sec. 2.3) holds up to the unlock time; after that, deviation is non-increasing.

## N  Decoded Examples

We provide additional response examples for MT-Bench. We randomly selected one question per category.

> **Question:** Compose an engaging travel blog post about a recent trip to Hawaii, highlighting cultural experiences and must-see attractions.

**Baseline** (score: 1)
Title: "Unveiling the Paradise: A Cultural Journey Through Hawaii"
Introduction:

**SURELOCK** (score: 3)
Title: "A Journey Through Paradise: Unveiling the Hidden Gems of Hawaii"
Introduction: As the sun set over the lush landscapes of Hawaii, I felt

Figure 8: Quality comparison of responses between Baseline vs. SURELOCK response. The example question uses the record from MT-bench with question id= 81. Responses are from LLaDA-8B-Instrut with KL threshold $\varepsilon = 5e - 4$, $N_{\text{gen}} = 128$, $S = 128$.

**Question:** Embrace the role of Sheldon from "The Big Bang Theory" as we delve into our conversation. Don't start with phrases like "As Sheldon". Let's kick things off with the following question: "What is your opinion on hand dryers?"

**Baseline** (score: 2)
Hand dryers are useless.

**SURELOCK** (score: 1)
Hand dryers.

Figure 9: Quality comparison of responses between Baseline vs. SURELOCK response. The example question uses the record from MT-bench with question id= 92. Responses are from LLaDA-8B-Instrut with KL threshold $\varepsilon = 5e - 4$, $N_{\text{gen}} = 128$, $S = 128$.

**Question:** Thomas is very healthy, but he has to go to the hospital every day. What could be the reasons?

**Baseline** (score: 3)
here could be several reasons why Thomas has to go to the hospital every day, despite being very healthy. Some possible reasons include:
1. Chronic illness: Thomas may have a chronic illness that requires daily medical attention, such as diabetes, cancer, or a heart condition.
2. Injury recovery: Thomas may be recovering from a serious injury or surgery that requires daily monitoring and care.
3. Medical treatment: Thomas may be undergoing a medical treatment that requires daily visits to the hospital, such as chemotherapy or radiation therapy.
4. Medical procedure: Thomas may be undergoing a medical procedure that requires daily visits to

**SURELOCK** (score: 4)
There could be several reasons why Thomas has to go to the hospital every day despite being healthy. Some possible reasons include:
1. Chronic illness: Thomas might have a chronic illness that requires daily medical attention. 2. Medical treatment: He could be undergoing a medical treatment that requires daily visits to the hospital. 3. Recovery period: Thomas might be recovering from an injury or surgery that requires daily monitoring and care. 4. Medical condition: He could be managing a medical condition that requires daily monitoring and treatment. 5. Medical procedure: Thomas might be undergoing a medical procedure that requires daily visits to the hospital.

Figure 10: Quality comparison of responses between Baseline vs. SURELOCK response. The example question uses the record from MT-bench with question id= 103. Responses are from LLaDA-8B-Instrut with KL threshold $\varepsilon = 5e - 4$, $N_{\text{gen}} = 128$, $S = 128$.

**Question:** When rolling two dice, what is the probability that you roll a total number that is at least 3?

**Baseline** (score: 2)
There are $6 \times 6 = 36$ possible outcomes when rolling two dice. We need to count the number of outcomes where the sum of the dice is at least 3.
The possible sums are 3, 4, 5, and 6.
- Sum of 3: 2 outcomes (1,2), (2,1) - Sum of 4: 3 outcomes (1,3), (2,2), (3,1) - Sum of 5: 4 outcomes (1,4), (2,3), (3,

**SURELOCK** (score: 3)
There are $6 \times 6 = 36$ possible outcomes when rolling two dice. We will count the number of outcomes where the sum is at least 3.
- Sum = 3: (1,2), (2,1) - Sum = 4: (1,3), (2,2), (3,1) - Sum = 5: (1,4), (2,3), (3,2), (4,1) - Sum = 6: (1,5), (2,4), (3,3), (4,

Figure 11: Quality comparison of responses between Baseline vs. SURELOCK response. The example question uses the record from MT-bench with question id= 114. Responses are from LLaDA-8B-Instrut with KL threshold $\varepsilon = 5e - 4$, $N_{\text{gen}} = 128$, $S = 128$.

**Question:** Write a function to find the highest common ancestor (not LCA) of two nodes in a binary tree.

**Baseline** (score: 3)
Here is a Python function to find the highest common ancestor (HCA) of two nodes in a binary tree:
"' class TreeNode: def __init__(self, val0, left=None, right=None): self.val = val self.left = left self.right = right
def find_hca(root, p, q): if root is None: return None if root == p or root == q: return root left = find_hca(root.left, p, q) right = find_hca(root.right,

**SURELOCK** (score: 3)
Here's a Python function to find the highest common ancestor (HCA) of two nodes in a binary tree:
"' class TreeNode: def __init__(self, val=0, left=None, right=None): self.val = val self.left = left self.right = right
def find_hca(root, p, q): if root is None: return None if root == p or root == q: return root left = find_hca(root.left, p, q) right = find_hca(root.right,

Figure 12: Quality comparison of responses between Baseline vs. SURELOCK response. The example question uses the record from MT-bench with question id= 125. Responses are from LLaDA-8B-Instrut with KL threshold $\varepsilon = 5e-4$, $N_{\text{gen}} = 128$, $S = 128$.

**Question:** Please read the paragraph below and count how many times the words "Amazon", "river", and "you" appear. Please present the results in the format of "word, number of appearances" with each word on a separate line. Sort the lines in order of the number of appearances. The Amazon, a mesmerizing expanse of nature's wonders, is home to the legendary Amazon River. Flowing through awe-inspiring landscapes like the Amazon rainforest, the river weaves its way through Brazil, Colombia, and Peru, giving life to countless creatures. From the mighty jaguars prowling the Amazon jungle to the vibrant macaws soaring above the canopy, this remarkable region teems with biodiversity. Deep within the river's currents, magnificent pink river dolphins gracefully glide alongside piranhas and electric eels. Along the riverbanks, you'll find bustling cities like Manaus, where the urban meets the wild, and Iquitos, a gateway to the heart of the Amazon rainforest. As you venture further, the Amazon River reveals hidden gems like the captivating Anavilhanas Archipelago, a mosaic of islands brimming with rare species. Embark on an adventure, explore the enchanting Amazon River, and immerse yourself in a world teeming with life and untamed beauty.

**Baseline** (score: 3)
Amazon, 2 river, 2 you, 2

**SURELOCK** (score: 3)
Amazon, 2 river, 2 you, 2

Figure 13: Quality comparison of responses between Baseline vs. SURELOCK response. The example question uses the record from MT-bench with question id= 136. Responses are from LLaDA-8B-Instrut with KL threshold $\varepsilon = 5e-4$, $N_{\text{gen}} = 128$, $S = 128$.

**Question:** The city of Vega intends to build a bridge that will span the Vegona River, covering a distance of 1.8 kilometers. The proposed location falls within a seismically active area that has experienced several high-magnitude earthquakes. Given these circumstances, what would be the best approach to constructing the bridge?

**Baseline** (score: 7)
Given the seismically active nature of the area, the best approach to constructing the bridge would be to use a flexible and resilient design that can withstand the forces of earthquakes. This could involve using a combination of materials such as steel, concrete, and timber, as well as incorporating features such as flexible joints and base isolators to allow the bridge to move in response to seismic activity. Additionally, the bridge should be designed to be lightweight and compact to minimize the risk of damage during an earthquake. Finally, the bridge should be designed and constructed according to seismic codes and regulations to ensure its safety and stability in the event of earthquakes.

**SURELOCK** (score: 7)
Given the seismically active nature of the area, the best approach to constructing the bridge would be to use a flexible and resilient design that can withstand the effects of earthquakes. This could include using a combination of concrete and steel, as well as incorporating seismic isolation systems such as base isolation or rubber bearings. Additionally, the bridge should be designed to be flexible and able to absorb and dissipate energy from seismic waves, rather than trying to resist them. It is also important to conduct thorough research and testing to evaluate the bridge's performance under various earthquake scenarios to ensure that the bridge remains safe and stable in the face of seismic activity.

Figure 14: Quality comparison of responses between Baseline vs. SURELOCK response. The example question uses the record from MT-bench with question id= 147. Responses are from LLaDA-8B-Instrut with KL threshold $\varepsilon = 5e - 4$, $N_{\text{gen}} = 128$, $S = 128$.

**Question:** Which methods did Socrates employ to challenge the prevailing thoughts of his time?

**Baseline** (score: 6)
Socrates employed several methods to challenge the prevailing thoughts of his time. One of the most famous methods was the Socratic method, which involved asking a series of questions to uncover the underlying assumptions and beliefs of his listeners. This method encouraged his listeners to think critically and question their own beliefs.
Another method Socrates used was the concept of the "good life," which he believed was the only way to achieve true happiness. He argued that the good life involved living in accordance with reason and virtue, rather than pursuing material wealth or power.
Finally, Socrates used the concept of the "just city" to challenge the

**SURELOCK** (score: 6)
Socrates employed several methods to challenge the prevailing thoughts of his time. One of the most famous methods was the Socratic method, which involved asking a series of questions to encourage someone to think critically and examine their own beliefs. This method was used to uncover the underlying assumptions and beliefs of those around him.
Another method Socrates used was the use of irony. He would often use irony to highlight the absurdity of certain beliefs or ideas, and to encourage others to question their own assumptions.
Finally, Socrates also used the power of dialogue to challenge the prevailing beliefs of his time. He would often engage in debates

Figure 15: Quality comparison of responses between Baseline vs. SURELOCK response. The example question uses the record from MT-bench with question id= 158. Responses are from LLaDA-8B-Instrut with KL threshold $\varepsilon = 5e - 4$, $N_{\text{gen}} = 128$, $S = 128$.

