# OpenReview forum: "Stopping Computation for Converged Tokens in Masked Diffusion-LM Decoding"
_ICLR.cc/2026/Conference — ICLR 2026 Poster_

### Official Review · Reviewer_kJZ5 · 2025-10-31

**Soundness:** 3
**Presentation:** 3
**Contribution:** 2
**Rating:** 4
**Confidence:** 3

**Summary:**

SureLock locks converged tokens during masked diffusion LM decoding, skips their query projection + FFN for the rest of the sampling steps, and just reuses cached K/V so other tokens can still attend to them. This is supposed to drop per-step cost from O(N2d) to O(MNd), where M is the number of still-active tokens. Authors report ~$30–50$% FLOP reduction on LLaDA-8B with comparable quality. Authors also demonstrate TPS gains relative to the baseline for large batches and longer generation.

Overall, I like the idea -- it is simple and intuitive,  but the contribution feels modest without wall-clock comparisons to prior methods that tackle the same problem -- inference latency. I rank this paper at 4 for now and open to raising my score if the authors add solid baselines (and ideally more shorter generation results).

**Strengths:**

- Simple idea (to its credit), training free
- Proposed method results in TPS  speed ups for large-batch/long-generation settings.
- The design-theoretic bound helps motivate KL as a convergence signal (even if not yet operationalized)

**Weaknesses:**

- Table 2, perplexity for WikiText-103: it seems that this method results in considerable perplexity drop even for 256 tokens length generation (~4-8%) and even more for shorter context.
- For low-batch, short-generation $(N <= 256)$ proposed method shows little to no wall-clock benefit.Note that $N=256$
is not very short (it’s a few sentences), so the absence of speedup here is concerning for interactive use.
- Despite method is positioned as orthogonal to existing diffusion-LM accelerators (e.g., KV-cache reuse/adaptive recompute, fewer denoising steps, slow/fast schedulers), I do believe that the experiments lack direct wall-clock comparisons to at least one prior work. This makes it hard to judge the contribution.

**Questions:**

- Do authors have quantitative results for the optional unlocking variant? (Appendix H)?
- Since a locked token cannot be revised, can you share a few cherry-picked bad cases where premature locking leads to noticeable quality degradation?

---

> ### Author Response · Authors · 2025-11-21
> **Response to Reviewer kJZ5 (Part 1)**
>
> We sincerely thank the reviewer for your dedication to carefully understanding our contributions and for recognizing the simplicity and intuition behind SureLock, as well as for being open to raising the score based on additional baselines.
>
> # Weakness-1: Gen.-PPL on Short Length Generation
> We agree that the degradation for very short outputs is more visible than for longer generations. In our experiments, however, we deliberately use a single global locking threshold and schedule across all configurations, rather than tuning hyperparameters per setting, in order to reveal the intrinsic behavior of SureLock across a wide range of regimes. This choice makes it clear that the method is particularly effective in compute-heavy settings (longer outputs and more reverse steps), while the relative impact of locking on perplexity is more pronounced on short sequences.
>
> To address the reviewer’s suggestion and to demonstrate that this degradation is not inherent to the method, we added a small-scale experiment that explicitly varies the locking threshold. For a fixed short $N_{\rm gen}$, we run a one-dimensional sweep over $\varepsilon$ and report how the compute–quality trade-off changes. As expected, lowering $\varepsilon$ reduces the perplexity gap while only modestly shrinking the efficiency, illustrating that, in practice, one can tune the locking aggressiveness to each specific scenario if desired. In other words, SureLock exposes a controllable compute–quality knob via $\varepsilon$, and Theorem 1 precisely formalizes how tightening $\varepsilon$ reduces the worst-case terminal error, so the degradation we observe at short lengths reflects a particular operating point on this trade-off curve rather than a fundamental inability of SureLock to preserve performance.
>
> In the revised manuscript, we included these sweep results as **Experiment-6, Table 5**, illustrating that the locking threshold can indeed be optimized in principle as Theorem 1.
>
> | $N_{\text{gen}}$ | steps | $\varepsilon$      | ↓ Gen.-PPL$_\text{base}$ | ↓ $\mathcal{F}_\text{base}$      | ↓ Gen.-PPL$_\text{prop}$        | ↓ $\mathcal{F}_\text{prop}$              |
> |------------------|-------|--------------------|---------------------------|----------------------------------|----------------------------------|------------------------------------------|
> | 64               | 64    | $5\times10^{-8}$   | 3.4813                    | $8.976\times 10^{11}$            | 4.0033 (1.14×)                   | $5.930\times 10^{11}$ (0.66×)            |
> | 64               | 64    | $5\times10^{-7}$   | 3.4813                    | $8.976\times 10^{11}$            | 4.3489 (1.25×)                   | $5.430\times 10^{11}$ (0.60×)            |
> | 64               | 64    | $5\times10^{-4}$   | 3.4813                    | $8.976\times 10^{11}$            | 4.5722 (1.31×)                   | $5.203\times 10^{11}$ (0.58×)            |
> | 64               | 64    | $5\times10^{-3}$   | 3.4813                    | $8.976\times 10^{11}$            | 4.7596 (1.37×)                   | $4.664\times 10^{11}$ (0.52×)            |
>
>
> **Table 5.** Generation quality of continuation sequences with and without SureLock on LLaDA-8B-Base on WikiText-103 for different values of $\varepsilon$.

---

> ### Author Response · Authors · 2025-11-21
> **Response to Reviewer kJZ5 (Part 2)**
>
> # Weakness-2: Gap between Runtime vs. FLOPs
> We agree that low-batch, short-generation scenarios are important for interactive use, and we appreciate the reviewer highlighting this regime.
>
> In our current implementation, however, SureLock is evaluated using a straightforward PyTorch-based decoder with off-the-shelf dense kernels, and (as we discussed in the Limitations) the wall-clock TPS is strongly influenced by implementation and hardware details that are orthogonal to the core algorithmic contribution. As a result, the reduction in arithmetic FLOPs does not always translate into clear TPS gains in the low-batch, short-generation regime, even though we already see substantial speedups in more compute-heavy settings.
>
> In the revised version, **In Appendix L**, we added a detailed discussion of the sources of this FLOPs-TPS gap (e.g., irregular cache accesses) and the potential for hardware-specific optimizations (such e.g., fused kernels) that would make the wall-clock gains more closely track the FLOPs reductions, even in interactive settings. Moreover, we clearly stated that our TPS evaluations should be interpreted as a lower bound from a naive implementation, and that our primary efficiency claims are in terms of algorithmic FLOPs, while we leave a full systems-level optimization of the low-batch, short-generation regime to future work.

---

> ### Author Response · Authors · 2025-11-21
> **Response to Reviewer kJZ5 (Part 3)**
>
> # Weakness-3: Wall-clock Comparison to Orthogonal Work
> We agree that direct wall-clock comparisons are helpful for judging the practical impact of our method, and we appreciate the reviewer’s suggestion. As discussed in Section 1, SureLock decides which already-unmasked token positions can be removed from all future computation, and thus shrinks the set of token positions on which existing accelerators operate. In this sense, SureLock is designed to be orthogonal and complementary to these approaches rather than a replacement.
>
> That said, it is still informative to show how SureLock behaves in combination with at least one representative prior method in wall-clock terms. We therefore add a small-scale experiment using a surrogate implementation of a selection-based method [Liu et al., 2025] that selects a fixed fraction $k$ = 0.8 of active tokens at each step. We report FLOPs and TPS under three settings: (i) Selection only, (ii) SureLock only, and (iii) their combination (selecting up to $kN$ tokens only among currently unlocked positions). For a focused comparison, we did not search over $k$ for optimal acceleration, so the results of the selection baseline should be interpreted as conservative; our goal here is to illustrate the relative behavior. Even under this simple setting, the results show that the combination yields additional speedups over either component alone at comparable quality, supporting our claim that SureLock is complementary to selection-based sparsification rather than competing on the same axis. We hope that this additional comparison will make the contribution easier to judge.
>
> In the revised version, we added this additional results as **Table 4** in **Experiment-5**.
>
> | SureLock | Selection (k = 0.8) | ↑ TPS_prop/TPS_base | ↓ F_prop/F_base |
> |----------|---------------------|----------------------------------------|------------------------------------------------------|
> | ✓        | –                   | 1.30×                                  | 0.54×                                               |
> | –        | ✓                   | 1.18×                                  | 0.64×                                               |
> | ✓        | ✓                   | **1.73×**                              | **0.28×**                                           |
>
> **Table 4.** Algorithmic FLOPs and runtime for different acceleration methods, evaluated on LLaDA-8B-Instruct on MT-Bench with \(N_{\text{gen}} = S = 256\).

---

> ### Author Response · Authors · 2025-11-21
> **Response to Reviewer kJZ5 (Part 4)**
>
> # Questions-1: Implementation of the Unlocking Variant
> We appreciate the reviewer’s interest in the optional unlocking variant in Appendix M.
>
> We do not currently have quantitative results for this variant; its purpose in the submission is to illustrate that our analysis in Theorem 1 naturally extends to a non-monotone rule in which previously locked positions can be unlocked under suitable conditions, rather than to present a second fully evaluated algorithm.
>
> Throughout the paper, we chose to focus on the simplest instantiation of SureLock (i.e., a monotone locking rule based on step-wise KL) **since, in the regimes we study, we did not observe practical failure modes** that would clearly require unlocking. Our main evaluation results show no consistent performance degradation across most configurations (Table 2 and 3). Moreover, as discussed in **our responses to Weaknesses 1**, the degradation on very short generations is a problem that can be solved in principle i.e., sliding $\varepsilon$.
>
> As an **additional support**, we see no degradation on a stricter downstream task, i.e., code generation on HumanEval benchmark, where correctness is judged by executing the generated code against several unit tests and even small local errors can cause failure. In a small-scale evaluation on the first 70 HumanEval problems under a representative configuration (LLaDA-8B-Instruct, $N_{\text{gen}}=S=256$), SureLock matches the baseline in Pass@1 while achieving roughly $0.5\times$ the algorithmic FLOPs.
>
> We included these results as **Table 6** in **Appendix J** in the revised manuscript.
>
> | $N_{\text{gen}}$ | steps | $\varepsilon$       | Pass@1$_\text{base}$ | $\mathcal{F}_\text{base}$       | Pass@1$_\text{prop}$       | $\mathcal{F}_\text{prop}$              |
> |------------------|-------|---------------------|-----------------------|----------------------------------|----------------------------|-----------------------------------------|
> | 256              | 256   | $5\times10^{-4}$    | 18.6                  | $3.648\times 10^{12}$           | $20.3\,(1.09\times)$       | $1.716\times 10^{12}\,(0.470\times)$   |
> | 256              | 256   | $5\times10^{-3}$    | 18.6                  | $3.648\times 10^{12}$           | $18.6\,(1.00\times)$       | $1.676\times 10^{12}\,(0.459\times)$   |
>
> **Table 6.** Quality changes in generated responses by LLaDA-8B-Instruct on HumanEval.
>
> # Questions-2: Existence of Corrupted Examples
>
> We appreciate the request for concrete failure cases. Under our default configurations, genuinely catastrophic examples are in fact very hard to find, which is consistent with the design of SureLock: locking is only applied to positions that have already been unmasked at relatively high confidence by the masked diffusion sampler (Algorithm 1), so locking typically results in small, local differences in wording rather than severe breakdowns of the output.
>
> In our targeted qualitative analysis, we **did not observe cases where locking drastically changes the topic, intent, or instruction-following**; instead, the differences are mostly minor paraphrasing or slightly less natural phrasing (Fig.4 and Appendix N). For example, the baseline produces the phrase “resilient design that can withstand the forces of earthquakes,” whereas SureLock yields “resilient design that can withstand the effects of earthquakes” in Figure 14, Appendix N.
>
> Furthermore, as discussed in our **response to Question 1**, we see no degradation on a stricter downstream task (i.e., code generation) where even small local errors can cause failure, which further supports that the surface-level differences introduced by locking rarely translate into semantic or structural failures in practice.

---

### Official Review · Reviewer_BReR · 2025-11-01

**Soundness:** 4
**Presentation:** 4
**Contribution:** 4
**Rating:** 6
**Confidence:** 2

**Summary:**

The paper proposes SureLock, a decoding-time method for masked diffusion language models (MDLMs) that permanently locks token positions once their posterior is locally "stable", thereby skipping Q-projection and FFN for those positions while caching K/V. This changes the dominant per-step cost from $O(N^2 T)$ to $O(MNT)$, where $M$ is the number of still-active positions. The paper establishes a local step-wise KL threshold (with optional confidence gating) for locking, and a theoretical bound for the terminal log-probability error. Experiments on LLaDA-8B (Base/Instruct) show 30–50% algorithmic FLOPs reduction with small changes in generation quality.

**Strengths:**

The study is very relevant and practical for discrete diffusion language model deployment.

The algorithm has simple, orthogonal develop towards temporal/reuse accelerations and integrates with K/V reuse and selective refresh. These are novel development in my opinion.

The algorithm is clearly presented and the paper is well written.

**Weaknesses:**

The bound in Theorem 1 relies on many assumptions, such as geometric tail contraction and Lipschitzness. These conditions are not carefully verified.

The base model shows non-trivial PPL degradation for short outputs ($N_{\rm gen}$ small). It may be interesting to propose adaptive locking schedules.

Empirical results are on LLaDA-8B Base/Instruct and two benchmarks. Evaluating long-context discrete diffusion models or other tasks would make the results more convincing and probe the robustness of the method.

**Questions:**

Can you estimate the constants in Assumptions (A2)-(A4) so that Theorem 1 becomes more explicit and interpretable?

The proposed locking is monotone, is it possible or beneficial to consider later free to "locked" tokens? This might matter for short length generations.

---

> ### Author Response · Authors · 2025-11-21
> **Response to Reviewer BReR (Part 1)**
>
> We sincerely thank the reviewer for recognizing that our acceleration strategy is orthogonal to the existing methods and for recognizing the novelty of our approach; we are very grateful for the time and care you invested in understanding these aspects of our work.
>
> # Weakness-1/Question-1: Strength of Assumptions
> We appreciate the reviewer’s careful reading of Theorem 1 and its assumptions. We agree that the bound is derived under non-trivial conditions such as geometric tail contraction and Lipschitz continuity, and that these assumptions are not carefully verified for the models used in our experiments. In hindsight, the current draft does not clearly articulate the strength and intended scope of these assumptions, which we acknowledge can be misleading.
>
> Our intention with Theorem 1 is to provide a stylized theoretical analysis that justifies the design of SureLock (i.e., using step-wise KL as the locking signal) rather than to give a numerically precise error prediction for a specific setting. Assumptions A1–A4 should be understood as simplifying regularity conditions introduced for analytical tractability, not as properties to be verified with tight constants on large deep models, as is common in theoretical studies.
>
> Among them, A2 may appear particularly strong; to reassure that it is not completely at odds with practice, **we will provide an empirical investigation (Appendix C, Figure 5)** showing that the step-wise KL between successive posteriors decays approximately geometrically in our setups, in line with the intended contractive behavior.
>
> In the revised manuscript, we **(i) explicitly stated in Section 2.3** that A1–A4 are such regularity assumptions and that Theorem 1 is meant as an idealized design justification for using step-wise KL as the locking signal, and **(ii) added a new figure in Appendix C** showing the step-wise KL decay across diffusion steps, as discussed above.
>
> We hope this clarifies the role of Theorem 1: it explains, in an idealized contractive regime, why locking errors do not explode, while our empirical results (Tables 2, 3, 5, 6,  and the new plots) indicate that this is indeed the regime in which we operate.

---

> ### Author Response · Authors · 2025-11-21
> **Response to Reviewer BReR (Part 2)**
>
> # Weakness-2: PPL degradation for short sequence generation
> We agree that the degradation for very short outputs (small $N_{\rm gen}$) is more visible than for longer generations. In our experiments, however, we deliberately use a single global locking threshold and schedule across all configurations, rather than tuning hyperparameters per setting, in order to reveal the intrinsic behavior of SureLock across a wide range of regimes. This choice makes it clear that the method is particularly effective in compute-heavy settings (longer outputs and more reverse steps), while the relative impact of locking on perplexity is more pronounced on short sequences.
>
>
> To address the reviewer’s suggestion and to demonstrate that this degradation is not inherent to the method, we add a small-scale experiment that explicitly varies the locking threshold. For a fixed short $N_{\rm gen}$, we run a one-dimensional sweep over $\varepsilon$ and report how the compute–quality trade-off changes. As expected, lowering $\varepsilon$ reduces the perplexity gap while only modestly shrinking the efficiency, illustrating that, in practice, one can tune the locking aggressiveness to each specific scenario if desired. In other words, SureLock exposes a controllable compute–quality knob via $\varepsilon$, and Theorem 1 precisely formalizes how tightening $\varepsilon$ reduces the worst-case terminal error, so the degradation we observe at short lengths reflects a particular operating point on this trade-off curve rather than a fundamental inability of SureLock to preserve performance.
>
> In the revised manuscript, we include these sweep results as **Table 5** in **Experiment-6**, illustrating that the locking threshold can indeed be optimized, exactly as suggested by Theorem 1, which characterizes how tightening $\varepsilon$ trades off worst-case terminal error against computational savings.
>
> | $N_{\text{gen}}$ | steps | $\varepsilon$      | ↓ Gen.-PPL$_\text{base}$ | ↓ $\mathcal{F}_\text{base}$      | ↓ Gen.-PPL$_\text{prop}$        | ↓ $\mathcal{F}_\text{prop}$              |
> |------------------|-------|--------------------|---------------------------|----------------------------------|----------------------------------|------------------------------------------|
> | 64               | 64    | $5\times10^{-8}$   | 3.4813                    | $8.976\times 10^{11}$            | 4.0033 (1.14×)                   | $5.930\times 10^{11}$ (0.66×)            |
> | 64               | 64    | $5\times10^{-7}$   | 3.4813                    | $8.976\times 10^{11}$            | 4.3489 (1.25×)                   | $5.430\times 10^{11}$ (0.60×)            |
> | 64               | 64    | $5\times10^{-4}$   | 3.4813                    | $8.976\times 10^{11}$            | 4.5722 (1.31×)                   | $5.203\times 10^{11}$ (0.58×)            |
> | 64               | 64    | $5\times10^{-3}$   | 3.4813                    | $8.976\times 10^{11}$            | 4.7596 (1.37×)                   | $4.664\times 10^{11}$ (0.52×)            |
>
> **Table 5.** Generation quality on LLaDA-8B-Base on WikiText-103 for different values of $\varepsilon$.

---

> ### Author Response · Authors · 2025-11-21
> **Response to Reviewer BReR (Part 3)**
>
> # Weakness-3: Evaluation on More Complex Benchmark
> We agree that our current experiments focus on basic tasks such as language modeling and instruction following. We chose these settings because they are standard proxies for open-ended generation quality and instruction adherence, but we acknowledge that additional results on more complex tasks would further strengthen the empirical validation of our method.
>
> To address this, we add a small-scale experiment on code generation using HumanEval [Chen et al., 2021] as a more challenging benchmark that stresses global coherence and functional correctness. Code generation is highly sensitive to local errors (e.g., a single token can break syntax or change the program logic), since performance is measured by executing the generated code in a sandbox environment and reporting the pass rate (pass@1). This makes HumanEval a stringent test of whether premature locking induces harmful failures. We evaluate SureLock on the first 70 HumanEval problems under a representative configuration (LLaDA-8B-Instruct, $N_{\text{gen}} = S = 256$, matching our main experiments). In this setting, we do not observe any degradation in pass@1 compared to the baseline, while achieving roughly $0.5\times$ the algorithmic FLOPs.
>
>
> In the revised version, we mentioned these additional results **at the end of Experiment-3** and provided further details in **Appendix J, Table 6**.
>
> | $N_{\text{gen}}$ | steps | $\varepsilon$       | Pass@1$_\text{base}$ | $\mathcal{F}_\text{base}$       | Pass@1$_\text{prop}$       | $\mathcal{F}_\text{prop}$              |
> |------------------|-------|---------------------|-----------------------|----------------------------------|----------------------------|-----------------------------------------|
> | 256              | 256   | $5\times10^{-4}$    | 18.6                  | $3.648\times 10^{12}$           | $20.3\,(1.09\times)$       | $1.716\times 10^{12}\,(0.470\times)$   |
> | 256              | 256   | $5\times10^{-3}$    | 18.6                  | $3.648\times 10^{12}$           | $18.6\,(1.00\times)$       | $1.676\times 10^{12}\,(0.459\times)$   |
>
> **Table 6.** Quality changes in generated responses by LLaDA-8B-Instruct on HumanEval.

---

> ### Author Response · Authors · 2025-11-21
> **Response to Reviewer BReR (Part 4)**
>
> # Questions-2: Can we exceed monotonous locking?
>
> We appreciate this suggestion. In this work we intentionally restrict ourselves to a monotone locking rule: once a position is locked, its hidden/KV representations are never updated again. We chose this design partly for simplicity, both algorithmically and in the analysis of Theorem 1, and partly because our experiments did not reveal practical failure modes that would clearly require unlocking.
>
>
> We agree, however, that allowing “unlocking’’ of previously locked positions is a natural extension, especially for short generations where the impact of an early locking decision could in principle be larger. In fact, Appendix M already discusses an optional variant of SureLock in which previously locked positions can be unlocked after a fixed number of steps if they become uncertain again, and we show that this variant naturally fits within the framework of Theorem 1.
>
>
> However, our empirical results do not indicate a strong need for this extension: as discussed in our response to Weakness 2, the degradation in Gen.-PPL for short sequence generations can be resolved in principle by tuning the locking threshold, and, as noted in our response to Weakness 3, we do not observe any degradation on code generation task where even small local errors are highly detrimental.
>
>
> For these reasons, we did not implement the unlocking variant in our experiments and instead present it in the appendix as a principled mechanism that could be explored in future work if stronger conservatism is required in other settings.

---

### Official Review · Reviewer_ZEiy · 2025-11-01

**Soundness:** 3
**Presentation:** 3
**Contribution:** 3
**Rating:** 6
**Confidence:** 3

**Summary:**

This paper proposes SURELOCK, an efficient sampling method for Masked Diffusion Language Models that reduces algorithmic FLOPs. The method is based on a simple yet effective idea: once tokens stabilize during the diffusion process, they can be "locked," allowing their query projection and feed-forward sublayers to be skipped. The locking criterion is determined using step-wise KL divergence. Experimental results demonstrate that SURELOCK reduces algorithmic FLOPs by up to 50% without compromising generation quality.

**Strengths:**

1. The proposed method is simple yet effective, with experimental results demonstrating a substantial reduction in FLOPs without compromising generation quality.

2. Theoretical and experimental analyses further confirm the soundness and robustness of the proposed method.

**Weaknesses:**

1. While the experiments demonstrate the proposed method's effectiveness, incorporating relevant baselines for comparison would further validate the soundness of this work.

2. The experiments are limited to basic tasks such as language modeling and instruction following. Results on more complex tasks would further validate the method's effectiveness.

**Questions:**

Masked Diffusion Language Models are already known for faster inference compared to autoregressive models. Is trading off generation quality for better inference efficiency a worthwhile choice for Masked Diffusion Language Models?

---

> ### Author Response · Authors · 2025-11-21
> **Response to Reviewer ZEiy (part 1)**
>
> We thank the reviewer for the effort you devoted to understanding the contributions of our paper. The phrase “simple yet effective” that you used in the Strengths section is especially gratifying for us. In the following, we provide responses to the Weaknesses and Questions you raised.
>
> # Weakness-1; relevant baselines to demonstrate effectiveness
> As we already discussed in Section 1 (paragraphs 2 and 4), temporal- and reuse-based approaches optimize the number of reverse steps or reuse past partial computation, while selection-based approaches reduce per-step cost by querying only a fixed fraction k (0 < k < 1) of the active positions A_t, bringing the per-step compute from O(N) down to O(kN). In all of these cases, however, the number of queried positions remains proportional to N at every step, so per-step compute does not vanish even in later reverse steps.
>
> Conceptually, SureLock optimizes a different axis. As explained in Section 1 (paragraph 4), our method answers a complementary question: which token positions can be removed from future computation entirely. By design, SureLock operates only on already-unmasked token positions; once such a position is locked, it is never updated again and can be permanently removed from the candidate set on which temporal/reuse/selection strategies operate. In other words, SureLock shrinks the set of positions on which these methods act, rather than competing with them on the same axis of optimization. This is why we view temporal/reuse/selection approaches as orthogonal to SureLock in terms of acceleration, rather than as direct baselines for a head-to-head comparison.
>
> While we therefore do not treat these existing methods as primary baselines, we agree that it is still informative to empirically confirm that SureLock is complementary to them. To this end, we implement a surrogate selection-based method that selects a fixed fraction k = 0.8 of active tokens at each step [Liu et al., 2025], and report FLOPs and TPS under three settings: selection only, SureLock only, and their combination (selecting kN tokens only among unlocked positions).
> **Table 4 in Experiment-5** in the revised version shows that the combination yields additional speedups over either component alone, supporting our claim that SureLock and selection-based approach cooperate.
>
> | SureLock | Selection (k = 0.8) | ↑ TPS_prop / TPS_base |  ↓ F_prop / F_base |
> |----------|---------------------|------------------------|-------------------|
> | ✓        | -                   | 1.30×                 | 0.54×             |
> | -        | ✓                   | 1.18×                 | 0.64×             |
> | ✓        | ✓                   | **1.73×**             | **0.28×**         |
>
> **Table 4.** Algorithmic FLOPs and runtime for different acceleration methods, evaluated on LLaDA-8B-Instruct on MT-Bench with $N_{\text{gen}} = S = 256$.
>
> # Weakness-2: Evaluation on More Complex Benchmark
> We agree that our current experiments focus on basic tasks such as language modeling and instruction following. We chose these settings because they are standard proxies for open-ended generation quality and instruction adherence, but we acknowledge that additional results on more complex tasks would further strengthen the empirical validation of our method.
>
> To confirm this, we added small-scale experiments on code generation using HumanEval [Chen et al 2021] as a more challenging setting that stresses global coherence and functional correctness. Code generation is highly sensitive to local errors (e.g., a single token can break syntax or change the program logic) since it is examined by running generated code in a sandbox environment to report pass rate (Pass@1). It makes it a stringent test of whether premature locking induces harmful failures. Under a representative configuration, SureLock achieves no performance drop with about 0.5× efficiency in FLOPs.
>
> We stated the results in **Experiment-3** and added the details as **Table 6** in **Appendix J**.
>
> | $N_{\text{gen}}$ | steps | $\varepsilon$ | Pass@1$_\text{base}$ | $\mathcal{F}_\text{base}$    | Pass@1$_\text{prop}$      | $\mathcal{F}_\text{prop}$        |
> |--------------------|-------|-----------------|-------------------------|---------------------------------|-----------------------------|----------------------------------------|
> | 256                | 256   |  $5\times10^{-4}$ | 18.6                   | $3.648\times 10^{12}$        | $20.3\,(1.09\times)$      | $1.716\times 10^{12}\,(0.470\times)$ |
> | 256                | 256   | $5\times10^{-3}$ | 18.6                   | $3.648\times 10^{12}$        | $18.6\,(1.00\times)$      | $1.676\times 10^{12}\,(0.459\times)$ |
>
> **Table 6.** Quality changes in generated responses by LLaDA-8B-Instruct on HumanEval.

---

> ### Author Response · Authors · 2025-11-21
> **Response to Reviewer ZEiy (part 2)**
>
> # Questions-1: Isn't Diffusion LM already fast?
> We believe this question is based on an important but incomplete view of the speed vs. quality trade-off for masked diffusion LMs. These models can indeed be faster than autoregressive (AR) models in very low-step regimes, but that reducing sampling steps typically comes with a noticeable loss in generation quality [Nie et al., NeurIPS 2025]. In practice, to match or approach the quality of same-scale AR LMs, diffusion LMs need many sampling steps, often operating in a regime comparable to "one step per token," where raw inference speed is no longer better than AR LMs.
>
> SureLock is explicitly aimed at this high-quality, inefficient regime. Our goal is to reduce computational complexity at each step, rather than reducing sampling steps (i.e., reducing quality). In this regime, SureLock provides significant acceleration with only minor quality degradation, as evidenced by Tables 2 and 3, and with theoretical guarantees from Theorem 1.

---

### Official Review · Reviewer_XG2i · 2025-11-03

**Soundness:** 4
**Presentation:** 3
**Contribution:** 3
**Rating:** 8
**Confidence:** 3

**Summary:**

This paper introduces SURELOCK, a novel method to accelerate the iterative decoding process of Masked Diffusion Language Models (MDLMs). The core problem addressed is the computational redundancy in standard MDLMs, where self-attention and feed-forward networks are recomputed for all token positions at every decoding step, even for tokens that have already converged. SURELOCK proposes to dynamically identify and "lock" these converged tokens. The locking criterion is based on the step-wise KL divergence of a token's posterior probability distribution falling below a certain threshold. Once a token position is locked, its Query and FFN computations are permanently skipped in subsequent steps, while its Key and Value vectors are cached to allow other active tokens to continue attending to it. This reduces the dominant per-iteration computational cost. The authors provide a rigorous theoretical analysis to justify their use of a local KL-divergence criterion, proving that it bounds the terminal log-probability error. Empirical evaluations on LLaDA-8B demonstrate that SURELOCK achieves a 30–50% reduction in algorithmic FLOPs while maintaining comparable generation quality on language modeling and instruction-following tasks.

**Strengths:**

1.  **Well-defined Problem and an Intuitive Solution:** The paper effectively identifies a clear source of inefficiency in MDLMs—the redundant computation for already stable tokens. The proposed solution, SURELOCK, is intuitive and directly targets this issue by progressively reducing the set of actively computed tokens.

2.  **Principled Algorithm Design with Theoretical Backing:** A key strength of this work is that it goes beyond a purely heuristic approach. The authors provide a theoretical justification (Theorem 1) for their locking criterion, linking the local, step-wise KL divergence to a bound on the final log-probability error. This analysis provides a solid rationale for the algorithm's design and increases confidence in its stability.

3.  **Solid Empirical Validation:** The experimental evaluation is well-conducted. The authors report both theoretical efficiency gains (algorithmic FLOPs) and practical runtime performance (TPS), which is a good practice. The method is tested on both a foundational language modeling task and a more practical instruction-following task. The use of strong external LLMs for quality evaluation (LLaMA-3, GPT-4o) adds credibility to the claims that generation quality is largely preserved.

**Weaknesses:**

While the paper is strong overall, there are a few areas where the analysis could be deepened or clarified.

1.  **Empirical Grounding of Theoretical Assumptions:** The theoretical analysis relies on several assumptions, particularly (A2) "Geometric tail contraction," which states that the KL divergence decays at a geometric rate. This is a fairly strong assumption. The paper would benefit from an empirical investigation into how well this assumption holds for the models and tasks tested. Without this, the practical applicability of the derived error bound remains somewhat abstract.

2.  **Gap between Algorithmic FLOPs and Wall-Clock Time:** The paper acknowledges that the significant reduction in algorithmic FLOPs does not always translate to a proportional reduction in wall-clock time, especially in non-compute-bound scenarios. While the analysis of this gap is appreciated, a more in-depth discussion on the specific implementation overheads (e.g., managing the cache, irregular memory access) and the potential for hardware-specific optimizations would make the work more impactful from a practical systems perspective.

**Questions:**

1. Your locking criterion is based on the stability of the posterior distribution. Does this "distributional stability" always coincide with "semantic stability"? For instance, could a token position's distribution remain stable (low KL divergence) while oscillating between two semantically similar but different tokens (e.g., "happy" and "joyful")? If so, would locking it prematurely harm the final semantic nuance of the generated text?

2. The experiments were conducted with a default temperature of 0. How does sampling temperature interact with the SURELOCK mechanism? Intuitively, a higher temperature would lead to flatter, less certain posterior distributions, potentially delaying the locking of tokens and reducing the efficiency gains. Have you investigated this trade-off, and is there an optimal interplay between temperature and the locking threshold ε?

3. Theorem 1 provides a bound on the final error. From a qualitative perspective, how do errors from a potentially premature lock manifest in the generated text? Do they tend to be localized to the area around the locked token, or can they cause cascading failures that affect the global coherence of the sequence? Understanding the typical failure modes would be very insightful.

4. The SURELOCK principle of locking converged elements seems generalizable to other modalities where diffusion models are used, such as image or audio generation. Have you considered the applicability of this method to continuous data domains? What would be the main challenges in adapting the KL-divergence-based criterion from discrete token posteriors to continuous data representations?

---

> ### Author Response · Authors · 2025-11-21
> **Response to Reviewer XG2i (Part 1)**
>
> We thank the reviewer for the detailed review. The points you highlighted under Strengths closely align with the main messages we intended to convey, and we are sincerely grateful for the time and effort you devoted to understanding our work. We are also very pleased with the Soundness score of 4. Below, we provide our responses to the Weaknesses and Questions you raised.
>
> # Weakness-1: Empirical Grounding of Theoretical Assumptions
> As you correctly pointed out, the assumption that the KL divergence at an arbitrary token position monotonically decreases over diffusion steps is, in principle, theoretical. To better understand how it behaves in practice, we visualized the step-wise evolution of the KL divergence. As a small-scale investigation, we used LLaDA-8B-Instruct with $\varepsilon=5e-4$, $N_\text{gen}=128$, and $S=128$ , on MT-Bench's 16 samples. We can see that the KL divergence indeed monotonically decreases as the sampling proceeds; we can infer that A2 is not merely a theoretical assumption, but oba that does not significantly deviate from the practical situation.
>
> In the revised version, we added these results in **Appendix C, Figure 5**, and cited that in Section 2.3 of the main text.
>
> # Weakness-2: Gap between Algorithmic FLOPs and Wall-Clock Time
> We thank the reviewer for pointing out the gap between algorithmic FLOP reductions and improvements in wall-clock time. We agree that a more concrete discussion of this phenomenon is crucial for understanding the practical impact of our method.
>
> In **Appendix L** in the revised version, we added the following points of discussions, by (i) clearly explaining the reasons behind the gap between real-time speedup and FLOPs reduction, and (ii) emphasizing that the method is compatible with future hardware-specific optimizations and is likely to benefit from them.
>
> ----
>
> In our method, token positions whose posterior has stabilized are locked: for these positions we skip all subsequent attention and FFN computations and instead reuse the K/V vectors cached at the time of locking. As reverse process proceeds, the number of active tokens decreases, so the algorithmic FLOPs decreases substantially. However, as the reviewer points out, this does not always translate into gains in wall-clock time. For example, the following sources of overhead become significant:
>
> - **Irregular memory accesses.** Reusing frozen tokens requires reading/writing Q/K/V at non-contiguous indices (A_t, L_t), leading to scattered, poorly coalesced accesses and making some regimes more bandwidth-bound.
>
> - **Large cache traffic**. Even when many tokens are skipped, substantial K/V data must still be moved between cache and compute kernels, so cache traffic can dominate latency.
>
> - **Index/mask management overhead**. Each diffusion step must update active/frozen index sets from posterior KL values (compute KL, threshold, build index arrays, update/apply cache); these fixed costs become non-negligible when the main compute per step is small.
>
> - **Under-utilized dense kernels.** When only a small fraction of tokens is active, generic dense attention/FFN kernels operate on narrow slices of the sequence, reducing thread/warp occupancy and hardware utilization and thus limiting speedups relative to the FLOP reductions.
>
> Regarding the potential for hardware-specific optimizations, more specialized implementations could further narrow the gap between FLOP reductions and latency improvements. A full systems-level exploration is beyond the scope of this work, but the following countermeasures are considered effective:
> - **Cache layout & compaction.** Maintain K/V caches so that active tokens occupy contiguous regions by periodically compacting them and tracking a logical to physical index map. This enables more coalesced memory accesses and higher effective bandwidth.
>
> - **Custom kernels for dynamic active sets.** Use fused GPU kernels that (i) gather active tokens, (ii) compute attention/FFN only for them, and (iii) write results back into a compact cache. This avoids many small kernel launches and reduces global-memory traffic, matching the sparsity pattern induced by SureLock.
>
> - **Overlap cache ops with compute.** Use asynchronous copies and prefetching to load K/V cache segments for step $t+1$ while computing attention and FFN for step $t$, overlapping memory traffic with computation and reducing effective cache-access latency.

---

> > ### Comment · Reviewer_XG2i · 2025-11-23
> > **Response to Authors**
> >
> > Thanks for the authors'response, which fully addressed my concerns. I will therefore maintain my positive score. Btw, we expect the release of the code.

---

> > > ### Author Response · Authors · 2025-11-26
> > > **Response to Discussion (Reviewer XG2i)**
> > >
> > > Thank you for the continuous engagement with our discussion.
> > > We very much appreciate it.
> > > Yes, we will publicly release the code upon paper acceptance as already stated in the paper.
> > >
> > > Best regards,
> > > Authors of Submission #25210

---

> ### Author Response · Authors · 2025-11-21
> **Response to Reviewer XG2i (Part 2)**
>
> # Questions-1: Can uncertainty and locking behavior coexist?
>
> We appreciate this question, as it touches on an important aspect of our locking criterion. Conceptually, we agree that “distributional stability’’ (low KL between successive posteriors) does not strictly imply “semantic stability’’: in principle, a position could maintain a step-wise stable distribution while probability mass oscillates between two near-synonymous tokens (e.g., “happy’’ vs. “joyful’’), still yielding a small KL. We view such cases as essentially benign, since committing to one of several paraphrases has little impact on the overall meaning.
>
> What would be more concerning is locking when the model is still undecided between semantically distant alternatives (e.g., “he’’ vs. “she’’, “yes’’ vs. “no’’). In practice, however, this appears rare given the combination of masked diffusion and our policy: standard masked diffusion schedulers unmask [MASK] positions in order of confidence, and \textsc{SureLock} only applies its KL-based test to already-unmasked positions whose posteriors remain stable across steps. By the time a position is both unmasked and locked, its posterior is typically concentrated on a small set of semantically consistent tokens, so residual variation is mostly between close paraphrases rather than contradictory options.
>
> Empirically, this matches our observations: across a range of generation lengths and numbers of reverse steps (Tables 2 and 3), we do not see systematic or severe degradation in language modeling or instruction-following metrics when enabling locking, suggesting that any semantic drift due to locking is negligible in practice. In the revised manuscript, we add a brief discussion of this issue in the **Limitations** and provide further discussion in **Appendix E**.
>
> # Questions-2: How are Locking and Temperature related?
> We thank the reviewer for raising this point. In our experiments we fix the sampling temperature to 0 to isolate the effect of SureLock and avoid confounding it with the usual diversity–quality trade-offs of temperature tuning. Importantly, however, the locking mechanism itself does not rely on a particular sampling temperature.
>
> In our implementation, the locking criterion (Sec. 2.2) is based on the model’s posterior before temperature scaling, i.e., using the distribution obtained from the raw logits (equivalently, a $\tau = 1$ softmax). The sampling temperature $\tau$, when non-zero, is applied only in the categorical sampling that materializes discrete tokens, not in the probability distributions used for the KL-based stability test. Consequently, for a fixed threshold $\varepsilon$, the set of positions judged “stable’’ by SureLock is independent of $\tau$: increasing $\tau$ changes the diversity of sampled outputs, but does not in itself delay or accelerate locking in our current design.
>
> One could, alternatively, choose to apply the same temperature scaling to the posteriors before computing KL, i.e., compute $\mathrm{KL}(p_t^{(\tau)} \| p_{t-1}^{(\tau)})$ instead of $\mathrm{KL}(p_t \| p_{t-1})$. Mathematically, this defines a $\tau$-dependent divergence that changes which parts of the distribution differences are emphasized (e.g., flattening posteriors makes KL smaller and effectively relaxes the notion of “stability’’). However, in practice this mainly reparameterizes the sensitivity of the test and plays a similar role to adjusting the threshold $\varepsilon$ itself. For clarity and simplicity, we therefore keep the KL computation temperature-agnostic ($\tau = 1$) and let $\varepsilon$ control how tolerant SureLock is to distributional changes, while $\tau$, if used, controls decoding diversity.
>
> In the revised version, we added clarification that SureLock is independent of sampling temperature and that this choice does not in itself delay or accelerate locking, in **Section 2.2 and Appendix F**.

---

> ### Author Response · Authors · 2025-11-21
> **Response to Reviewer XG2i (Part 3)**
>
> # Questions-3: How do locking errors manifest on the surface?
> In SureLock, once a token position is unmasked and then locked, we stop updating its K/V representations and reuse the cached values in all subsequent steps. Any approximation error at that position can therefore influence later tokens that attend to it, and this influence can, in theory, propagate through attention all the way to the entire sequence.
>
> However, our qualitative inspection suggests that such effects remain localized in practice. Differences attributable to locking almost always appear as small changes in word choice or local phrasing around the affected region, rather than as global breakdowns of topic, discourse structure, or instruction-following. For example, the baseline produces the phrase “resilient design that can withstand the forces of earthquakes,” whereas SureLock yields “resilient design that can withstand the effects of earthquakes”; **Appendix N** shows similar minor, localized differences in our case studies. This is consistent with the quantitative results in Tables 2 and 3: across a variety of generation lengths, numbers of diffusion steps, and tasks, SureLock matches the baseline without systematic or severe degradation, indicating that errors induced by premature locking do not typically lead to significant cascading failures in global coherence.
>
> To further probe how severe the impact of these localized differences can be, we also evaluate SureLock on the HumanEval code-generation benchmark. HumanEval measures functional correctness by executing the generated code, so even small local errors can easily break syntax or change program semantics and thus have a large effect on task performance. In practice, we observe that SureLock matches the baseline on HumanEval (pass@1) while providing acceleration, indicating that the minor surface-level differences induced by locking rarely translate into syntactic failures or semantic misbehavior in this more stringent setting.
>
> In the revised version, we explicit stated that our qualitative case studies exhibit only small, localized differences (Sec. 3.2 Example Analysis), and added code-generation benchmarking as **Table 6** in **Appendix J.**
>
> | $N_{\text{gen}}$ | steps | $\varepsilon$ | Pass@1$_\text{base}$ | $\mathcal{F}_\text{base}$    | Pass@1$_\text{prop}$      | $\mathcal{F}_\text{prop}$           |
> |--------------------|-------|-----------------|-------------------------|---------------------------------|-----------------------------|----------------------------------------|
> | 256                | 256   | $5\times10^{-4}$ | 18.6                   | $3.648\times 10^{12}$        | $20.3\,(1.09\times)$      | $1.716\times 10^{12}\,(0.470\times)$ |
> | 256                | 256   | $5\times10^{-3}$ | 18.6                   | $3.648\times 10^{12}$        | $18.6\,(1.00\times)$      | $1.676\times 10^{12}\,(0.459\times)$ |
>
> **Table 6.** Quality changes in generated responses by LLaDA-8B-Instruct on HumanEval.
>
>
> # Questions-4: Applicability to Continuous Diffusion and Its Challenges
> We appreciate this question and agree that the basic “converge-then-lock’’ idea behind SureLock is, at a conceptual level, could be applied to continuous data e.g., images or audio.
>
> However, our current formulation and Theorem 1 are explicitly KL-based and rely on having a discrete local posterior (the softmax) at each position. In continuous diffusion, the reverse process is usually parameterized via point estimates rather than explicit per-location distributions, so there is no ready-made local posterior to plug into a KL-based stability test. At a heuristic level, one could instead define a converge-then-lock rule using distances between latent vectors across steps, but this falls outside the KL/posterior framework that underlies Theorem 1 and would require a separate formulation and analysis.
>
> We therefore view extending SureLock to continuous diffusion as a promising but non-trivial direction for future work. We briefly added these statements in **Limitations** in the revised manuscript.

---

### Author Response · Authors · 2025-11-21
**General Response**

We would like to thank all reviewers for their careful reading and for the many constructive comments and suggestions.
We have responded to each of the raised Weaknesses and Questions in separate official comments, addressing the points in a point-by-point manner.

We have also uploaded a revised version of the manuscript that incorporates these suggestions, including additional experiments, clarifications of our assumptions and theoretical scope, and expanded discussion of limitations and potential extensions.

Please note that, due to the added material, some figure/table/section numbers have changed in the revised manuscript compared to the original submission.

We hope that the revisions make the contribution and intended usage of SureLock clearer.

---

> ### Author Response · Authors · 2025-11-26
> **Minor Update: Highlighted Revisions**
>
> We have made a very minor update to the revised version of the manuscript.
>
> Specifically, **we have highlighted in blue the main changes** that were introduced during the rebuttal phase. We hope this makes it easier to check how the rebuttal has been reflected in the paper. Aside from this highlighting, there is no substantive difference from the previously uploaded revised version.
>
> Moreover, **we would be grateful if reviewers could briefly check the rebuttal responses and the revised manuscript**, to verify that their comments and suggestions have been appropriately addressed.
>
> Best regards,
> Authors of Submission #25210

---

### Author Response · Authors · 2025-12-02
**Official Comment to AC, SAC, and PCs**

Dear AC, SAC, and PCs,

We are very grateful for the time and care the committee and reviewers devoted to our submission, as well as for their positive overall ratings (8/6/6/4). The reviews highlight both the intuitive, theoretically grounded design and the solid empirical support.

We interpreted the points raised under Weaknesses and Questions as opportunities to better characterize the properties and limitations of our method, SureLock.

In our revision, we therefore:

- (**Theory**) clarified the role and intended strength of our theoretical assumptions and conducted an empirical investigation of a key convergence assumption,
- (**Experiments**) added further experiments on more challenging tasks (i.e., code generation) and short sequence lengths, showing that the method can be tuned in each regime, in line with our theoretical analysis, to achieve favorable combinations of efficiency and generation quality,
- (**Orthogonality**) demonstrated its practical orthogonality by combining SureLock with a representative existing approach and observing additional speedups beyond either component alone, and
- (**Systems**) added a detailed discussion of how reductions in FLOPs translate into wall-clock speedups, including the potential of hardware-level optimizations (e.g., fused kernels).

The additional evidence we provide does not change our main claims; instead, it reinforces them and clarifies the method's scope and limitations. (Reference: during the discussion phase, the participating reviewer also noted that this additional evidence fully addressed their earlier concerns.)

We again sincerely thank the reviewers and the committee for helping us improve the paper.


Best regards,

Authors of Submission #25210

---

### Meta-Review · Area_Chair_eKwG · 2026-01-06

**Summary:**

This paper is on the fast inference of diffusion LLMs. One major bottleneck of dLLM is that each forward pass requires evaluating every token, including those that have already been unmasked. This results in a substantial waste in compute. To address this, the authors propose to stop updating the attention blocks of certain unmasked tokens that meet a criteria on KL divergence. Once locked, the corresponding K/V values are cached and can be used in the attention computation of other tokens. The method resembles other K/V cache methods for dLLMs developed recently.

**Reviewer Concerns:**

All the reviewers agree the overall idea is simple but effective. The training-free nature makes it a plug-in for many existing models. One major concern is on some assumptions used to develop this method. They may not hold in practice. The reviewers suggest to provide some empirical evidence to support these assumptions. On the experiment side, the paper misses some baselines, and tasks considered could be too simple. The paper could benefit from adding more comprehensive experiments. The authors made efforts to address these comments in the rebuttal.

**Reviewer Scores:**

no

---

### Decision · Program_Chairs · 2026-01-26

Accept (Poster)